# Empirical Comparison of Membership Inference Attacks in Deep Transfer Learning

**Yuxuan Bai**                                                        *yuxuan.bai@helsinki.fi*
*Department of Computer Science*
*University of Helsinki*

**Gauri Pradhan**                                                *gauri.pradhan@helsinki.fi*
*Department of Computer Science*
*University of Helsinki*

**Marlon Tobaben**                                          *marlon.tobaben@helsinki.fi*
*Department of Computer Science*
*University of Helsinki*

**Antti Honkela**                                               *antti.honkela@helsinki.fi*
*Department of Computer Science*
*University of Helsinki*

**Reviewed on OpenReview:** *https://openreview.net/forum?id=UligTUCgdt*

## Abstract

With the emergence of powerful large-scale foundation models, the training paradigm is increasingly shifting from from-scratch training to transfer learning. This enables high utility training with small, domain-specific datasets typical in sensitive applications. Membership inference attacks (MIAs) provide an empirical estimate of the privacy leakage by machine learning models. Yet, prior assessments of MIAs against models fine-tuned with transfer learning rely on a small subset of possible attacks. We address this by comparing performance of diverse MIAs in transfer learning settings to help practitioners identify the most efficient attacks for privacy risk evaluation. We find that attack efficacy decreases with the increase in training data for score-based MIAs. We find that there is no one MIA which captures all privacy risks in models trained with transfer learning. While the Likelihood Ratio Attack (LiRA) demonstrates superior performance across most experimental scenarios, the Inverse Hessian Attack (IHA) proves to be more effective against models fine-tuned on PatchCamelyon dataset in high data regime.

## 1  Introduction

As foundation models increasingly power modern AI systems, their adaptation through transfer learning raises privacy concerns. Recent research has demonstrated that fine-tuned models can inadvertently memorize their training data rather than learning generalizable patterns (Chu et al., 2025), creating potential privacy vulnerabilities.

Membership inference attacks (MIAs) (Shokri et al., 2017) have emerged as a critical tool for quantifying such privacy leakage by determining whether specific data points were used during model training. These attacks not only provide empirical lower bounds on privacy guarantees of a training algorithm, but also expose privacy vulnerabilities in model training strategies. Despite significant advances in MIA methodologies, their evaluation has predominantly focused on models trained from scratch. Figure 1 shows the varying privacy vulnerabilities exploitable by MIAs between fine-tuned and from-scratch trained models, even when both are

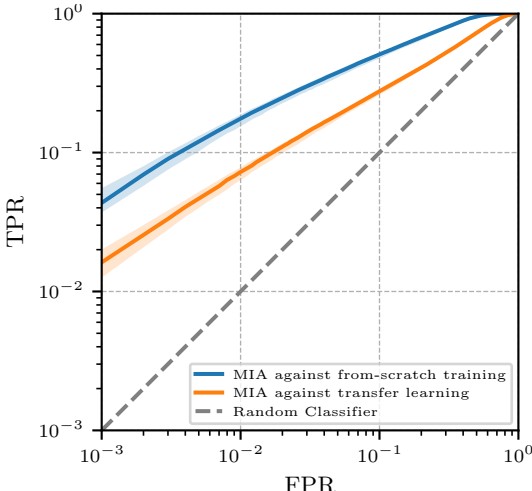

Figure 1: MIA efficacy as measured using LiRA (Carlini et al., 2022) against WideResNet-50-2 (Zagoruyko & Komodakis, 2016) trained from scratch using CIFAR-10 versus the same model pre-trained on ImageNet-1k (Deng et al., 2009) when only the last linear layer of the model is fine-tuned on CIFAR-10. The results are averaged over 3 repeats, with each repeat using $M + 1$ target models ($M = 64$) that share the same optimized hyperparameters obtained through hyperparameter optimization (HPO). The errorbars represent the interquartile range (IQR) of corresponding TPR at FPR. The plot demonstrates that the attack does not behave similarly across the 2 training paradigms, highlighting the need to investigate the performance of different MIA approaches against foundation models fine-tuned using deep transfer learning to ensure that a strong attack is used to evaluate their privacy risks.

trained on identical datasets. This divergence suggests that MIA efficacy in transfer learning fundamentally differs from that in from-scratch training.

Earlier works (Carlini et al., 2019; 2021; Lee et al., 2022; Kandpal et al., 2022) mainly focused on examining memorization of pre-training data by large foundation models. Meeus et al. (2025) recommend using fine-tuned versions of large language models to evaluate memorization using MIAs but do not conduct any experiments in this setting. Other studies using MIAs in transfer learning (Tobaben et al., 2023; 2024; Pradhan et al., 2025) have typically used a limited selection of attacks. A related study on the efficacy of MIAs against machine unlearning by Hayes et al. (2025) shows that the use of weaker versions of MIA overestimates the privacy protection provided by existing unlearning techniques. Thus, it is crucial to establish the relative efficacy of different MIAs to ensure that weaker attacks, which underestimate privacy risks, are not used to evaluate membership privacy in transfer learning scenarios.

**Our Contributions** In this work, we conducted a systematic evaluation of existing score-based MIAs (Yeom et al., 2018; Salem et al., 2019; Ye et al., 2022; Liu et al., 2022; Bertran et al., 2023; Li et al., 2024; Suri et al., 2024) in transfer learning context.

Motivated by the work by Tobaben et al. (2024), we investigate the relationship between MIA efficacy and fine-tuning dataset size using consistent experimental setups. Our results confirm that MIA efficacy generally decreases as the number of examples per class increases for most score-based attacks in transfer learning, consistent with the power-law relationship previously observed. However, we identify a notable exception: the white-box Inverse Hessian Attack (IHA) (Suri et al., 2024) exhibits markedly different behavior, demonstrating superior performance in high-shot regimes on PatchCamelyon compared to black-box methods. Additionally, we analyze the effects of changing the training paradigm and properties of the attacks on MIA efficacy.

## 2 Related Work

**Deep Transfer Learning**  Deep transfer learning has been widely adopted in machine learning, leveraging knowledge from source tasks to enhance performance on target tasks with limited data (Yosinski et al., 2014). The process involves pre-training on large-scale datasets to learn general-purpose feature representations, followed by fine-tuning on smaller, task-specific datasets, reducing data requirements and computational costs. However, this approach introduces privacy vulnerabilities. Models may memorize patterns from source datasets (Tramèr et al., 2024). Additionally, fine-tuning on small target datasets often leads to overfitting, increasing vulnerability to privacy attacks that can extract information about individual training samples. Researchers commonly use pre-trained models like ResNet (He et al., 2016; Kolesnikov et al., 2020) and Vision Transformer (ViT) (Dosovitskiy et al., 2021) due to computational constraints. Therefore, evaluating and mitigating privacy leakage during fine-tuning on sensitive downstream tasks forms a practical motivation for privacy research.

**Membership Inference Attacks**  Membership inference attacks (MIAs) aim to determine whether a specific data sample was used in training dataset of a target model. These attacks exploit differences in model behavior when responding to samples used for training the model (member samples) versus non-member samples, thereby compromising the privacy of sensitive data. MIAs are typically categorized based on the adversary's knowledge and access to the target model (Hu et al., 2022b). In the white-box setting, attackers have full access to the model's learned parameters, gradients, and architecture details. In contrast, black-box attacks operate with limited information, typically requiring knowledge of the data distribution and potentially access to model architecture and hyperparameters. Black-box MIAs further diverge into 2 primary variants: *score-based MIAs* that exploit the model's confidence scores, and *label-only MIAs* that function only with the predicted class labels (Li & Zhang, 2021; Choquette-Choo et al., 2021; Peng et al., 2024). Our work mainly focuses on score-based MIAs, as they represent the most potent yet practically feasible attacks.

**Shadow-model-based vs. Shadow-model-free MIAs**  Score-based MIAs can be further divided into shadow-model-based and shadow-model-free approaches. Shadow-model-based MIAs depend on *shadow training* (Shokri et al., 2017), a technique where the attacker trains surrogate models that mimic the behavior of the target model. These include methods such as ML-Leaks (Adversary 1) (Salem et al., 2019), Trajectory-MIA (Liu et al., 2022), Sequential-Metric based MIA (SeqMIA) (Li et al., 2024), Likelihood Ratio Attack (LiRA) (Carlini et al., 2022), and Robust MIA (RMIA) (Zarifzadeh et al., 2024). Despite their effectiveness, shadow-model-based MIAs require substantial computational resources, especially since their attack efficiency relies on training additional models. This limitation has motivated the development of more computationally efficient shadow-model-free alternatives, including LOSS attack (Yeom et al., 2018), Attack-P (Ye et al., 2022), and quantile-MIA (QMIA) (Bertran et al., 2023).

**Standard MIA Threat Model**  In the standard MIA threat model, formulated by Sablayrolles et al. (2019), the attacker is assumed to know the data distribution and the specifications of the model including the training procedure, model architecture, training hyperparameters, etc. Most of the attacks listed in Table 1 follow this threat model, except the Inverse Hessian Attack (IHA) proposed by Suri et al. (2024). IHA relies on 2 key assumptions: for a given target sample, it assumes knowledge of all $n-1$ records in a $n$-sized training dataset except for the target sample, and access to the target model's parameters (see Section 3.3 in Suri et al. (2024)). Thus, IHA does not follow the standard threat model for MIAs. Table 1 summarizes the threat models for all MIAs used in this paper.

## 3 Score-based MIAs

In this section, we describe the different MIAs employed in this paper and highlight how they differ in their approach to estimate membership privacy.

**Preliminaries**  Let $\mathcal{D}$ be a dataset sampled from data distribution $\pi$. This dataset is used to train a machine learning model $\mathcal{M}$ with parameters $\theta$. Next, we establish the probability notations used in the

Table 1: Summarizing MIAs in terms of the auxiliary information about the target model available to the attacker. Training Data Access implies that the attacker can access all but the target record in the training dataset.

| Attack | Target Model Access | | | | |
|---|---|---|---|---|---|
| | Data Distribution | Architecture | Hyperparameters | Model Parameters | Training Data Access |
| LOSS (Yeom et al., 2018) | ✓ | - | - | - | - |
| Attack-P (Ye et al., 2022) | ✓ | - | - | - | - |
| QMIA (Bertran et al., 2023) | ✓ | - | - | - | - |
| LiRA (Carlini et al., 2022) | ✓ | ✓ | ✓ | - | - |
| RMIA (Zarifzadeh et al., 2024) | ✓ | ✓ | ✓ | - | - |
| ML-Leaks (Salem et al., 2019) | ✓ | ✓ | ✓ | - | - |
| Trajectory-MIA (Liu et al., 2022) | ✓ | ✓ | ✓ | - | - |
| IHA (Suri et al., 2024) | ✓ | ✓ | ✓ | ✓ | ✓ |

paper. $\Pr(\theta|x)$ denotes the probability of observing parameters $\theta$ when $x$ in included in the training set, while $\Pr(\theta|\overline{x})$ denotes the probability of observing $\theta$ if $x$ is *not* in the training set. Conversely, $\Pr(x|\theta)$ represents the probability that $x$ was part of the training set that produced $\theta$.

### 3.1 Shadow-model-based MIAs

**ML-Leaks**  ML-Leaks (Adversary 1) (Salem et al., 2019) refines the shadow training approach (Shokri et al., 2017). It begins by training a shadow model using the dataset $\mathcal{D}_{\text{shadow}}$ sampled from $\pi$. However, instead of using the full prediction vector, ML-Leaks extracts only the top 3 posterior probabilities (or top 2 for binary-class datasets), ordered from highest to lowest for each sample in $\mathcal{D}_{\text{shadow}}$. Following this, it uses the trimmed probability vectors as inputs to train the attack model. This attack model can then be deployed to compute the membership scores for samples in $\mathcal{D}$.

**LiRA**  LiRA is a hypothesis testing framework for membership inference proposed by Carlini et al. (2022). For a given target model, it trains $M$ shadow models, such that the target sample, $x$, is included in the training dataset for $1/2$ of them (IN models) whereas it is excluded from the training dataset of the remaining $M/2$ models (OUT models). Using the predicted and logit-scaled confidence scores from these IN and OUT shadow models, the attacker can build the IN and OUT Gaussian distributions. Following this, the attacker can employ a likelihood ratio (LR) test (Neyman & Pearson, 1933) to compare $\Pr(\theta|x)$ against $\Pr(\theta|\overline{x})$:

$$\text{LR}_\theta(x) = \frac{\Pr(\theta|x)}{\Pr(\theta|\overline{x})}. \tag{1}$$

The attacker can use $\text{LR}_\theta(x)$ as the membership score to train a binary classifier to differentiate between member and non-member samples.

**Trajectory-MIA**  Liu et al. (2022) proposed Trajectory-MIA as an efficient alternative to LiRA. This is because Trajectory-MIA uses a single shadow model compared to LiRA's multiple shadow models' based approach. The method leverages knowledge distillation (Hinton et al., 2015) to simulate the target model's training trajectory. Specifically, it performs distillation on both the target and shadow models, minimizing Kullback-Leibler (KL) divergence between the student and teacher models' outputs. By recording the per-example training loss trajectory across distillation epochs, Trajectory-MIA captures temporal patterns that differ between member and non-member samples. These loss trajectories serve as feature vectors for a MLP classifier to predict memberships.

**RMIA**  RMIA (Zarifzadeh et al., 2024) further refines LiRA by incorporating knowledge about the population data $z \sim \pi$ in the likelihood ratio. It introduces a pairwise LR test that explicitly incorporates population samples $z$, computing the probability that the pairwise LR exceeds a preset threshold $\gamma$:

$$\Pr_{z \sim \pi}(\text{LR}_\theta(x, z) \geq \gamma) = \Pr_{z \sim \pi}\left(\frac{\Pr(\theta|x)}{\Pr(\theta|z)} \geq \gamma\right). \tag{2}$$

This pairwise LR formulation captures the relative relationship between a potential member sample $x$ and known non-member samples $z$ drawn from the population. By composing these pairwise comparisons, Zarifzadeh et al. (2024) contend that RMIA achieves greater robustness to distribution shifts between members and non-members.

### 3.2 Shadow-model-free MIAs

**LOSS Attack/ Attack-P**  LOSS attack (Yeom et al., 2018) uses loss on the target sample $\ell(\mathcal{M}(x), y)$ as a membership signal. Since the objective of training a machine learning model is usually to minimize their loss on the training samples, it compares the loss on the target sample against a fixed threshold $\tau$ to infer membership. Attack-P (Ye et al., 2022) is an improvised version of the LOSS attack, which constructs an empirical cumulative distribution function (CDF) from the population samples. Unlike the LOSS attack which uses a fixed threshold $\tau$, Attack-P compares the loss of the target sample to the distribution of losses from known non-members, calculating what percentage of non-member losses fall below the target's loss. Specifically, these attacks compare $\Pr(x|\theta)$ with $\Pr(z|\theta)$ as the threshold to determine the membership of $x$.

**QMIA**  QMIA (Bertran et al., 2023) determines per-sample thresholds by performing quantile regression on the distribution of confidence scores obtained from known non-member data. By training a regression model with pinball loss to predict these thresholds at a desired false positive rate, QMIA creates a nuanced decision boundary that adapts to individual sample. This model-agnostic approach was proposed as an effective alternative to shadow-model-based MIAs in black-box settings where only API access is available.

**IHA**  Proposed by Suri et al. (2024), the inverse hessian attack (IHA) builds on recent advancements in discrete-time SGD dynamics (Liu et al., 2021; Ziyin et al., 2022). They show that optimal membership inference requires white-box access to the model parameters post-training and knowledge of all but the target record in the training dataset, rather than relying only on output predictions. IHA relies on a local similarity assumption, which posits that models trained with or without a specific data point converge to similar local minima. Under this assumption, the Hessian matrices at the respective optima share similar structure: $\mathbf{H}_* = \mathbf{H}_0(\boldsymbol{w}_0^*) = \mathbf{H}_1(\boldsymbol{w}_1^*)$, where $\boldsymbol{w}_0^*$ and $\boldsymbol{w}_1^*$ represent the optimal parameters for models trained without and with the target sample, respectively. In addition, the loss functions achieve similar values at these local minima: $L_* = L_0(\boldsymbol{w}_0^*) = L_1(\boldsymbol{w}_1^*)$. This assumption allows IHA to approximate the optimal membership inference by formulating the MIA scoring function using terms dependent on gradients and model parameters.

## 4 Experimental Setup

**Datasets**  We use CIFAR-10, CIFAR-100 (Krizhevsky, 2009), and PatchCamelyon (Veeling et al., 2018) in our experiments. CIFAR-10 and CIFAR-100 are common benchmark datasets for MIA evaluation. PatchCamelyon, including only 2 classes, enables experiments with substantially larger number of shots $S$ (examples per class), providing greater insight into how training set size affects MIA efficacy.

**Models**  We use ViT-B/16 (Dosovitskiy et al., 2021) and BiT-M-R50x1 (R-50) (Kolesnikov et al., 2020) as the backbone models for fine-tuning, both pre-trained on ImageNet-21k (Deng et al., 2009).

**Parameterization**  We employ 3 schemes for parameterization: (*i) Head-only*, where only the classification layer is replaced by a trainable linear layer, with initial weights set to 0, while the feature extraction backbone remains frozen; (*ii) ALL*, where all parameters are trainable during fine-tuning, with initialization from the pre-trained weights; and (*iii) FiLM*, where FiLM adapters (Perez et al., 2018) are introduced throughout the network alongside a trainable classification head. This parameter-efficient technique, applicable to both convolutional and transformer architectures, enables more expressive adaptation while minimizing trainable parameters compared to full fine-tuning. Although alternatives such as LoRA (Hu et al., 2022a), and CaSE (Patacchiola et al., 2022) exist, FiLM is selected due to its demonstrated effectiveness in parameter-efficient few-shot transfer learning (Shysheya et al., 2023; Tobaben et al., 2023).

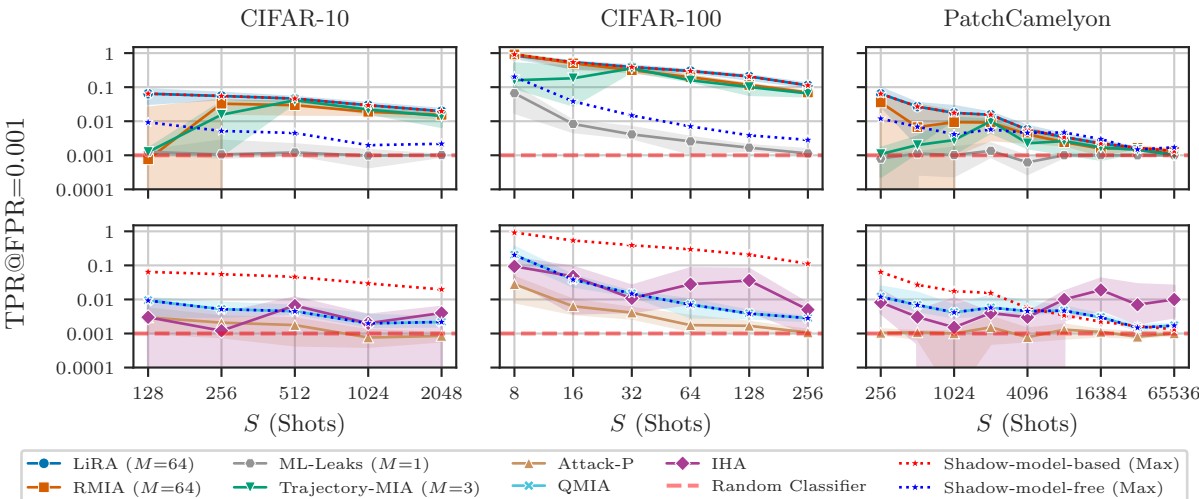

Figure 2: **MIA efficacy against ViT-B/16 (Head-only) models as a function of $S$ (shots).** Upper: Shadow-model-based attacks using $M$ shadow models. Lower: Shadow-model-free attacks. The errorbars represent the interquartile range (IQR) of the estimated TPR at fixed FPR and the dotted lines represent the maximum of the median MIA efficacy of shadow-model-based and black-box shadow-model-free attacks (IHA excluded). Shadow-model-based attacks generally demonstrate more stable and stronger MIA efficacy compared to shadow-model-free attacks. In the high-shot regime of PatchCamelyon, however, the white-box IHA has a considerable advantage over other MIAs in terms of MIA efficacy, leveraging on its access to all but the target record in the training dataset. Results are averaged over 10 repeats and we use 1 target model per repeat.

**Hyperparameter Optimization**   Before model training, we perform hyperparameter optimization (HPO) to identify the optimal set of hyperparameters to train the target model. We use the same set of hyperparameters to train both the target and the shadow model(s). We begin by sampling $1/2$ of the training dataset ($\mathcal{D}$) for HPO. In each HPO trial, we use 70% of the data for training the model while the remaining 30% is used as validation dataset. We run the HPO for 20 trials to explore the hyperparameter space. We implement HPO using Optuna (Akiba et al., 2019) with Tree-structured Parzen Estimator (TPE) algorithm (Bergstra et al., 2011). Table 2 summarizes the hyperparameters and their corresponding search ranges used in our experiments.

Table 2: Hyperparameter search ranges used for Bayesian optimization with Optuna.

| Hyperparameter | Parameterization | Range |
|:---:|:---:|:---:|
| Epoch | Head-only | $[1, 200]$ |
| | FiLM | $[1, 40]$ |
| Train Batch Size | Head-only/FiLM | $[10, 1000]$ |
| Learning Rate | Head-only/FiLM | $[10^{-7}, 10^{-2}]$ |

**Metrics**   The metrics used to evaluate the efficacy of MIAs include: (*i*) *log-scaled ROC* curves to visualize the trade-off between true positive rate (TPR) and false positive rate (FPR), with emphasis on the low FPR region, (*ii*) *TPR at low FPR* to measure MIA efficacy when the false positives must be minimized, better representing realistic attack scenarios (Carlini et al., 2022), and (*iii*) *Interquartile Range (IQR)* serving as a statistical measure to quantify uncertainty. It is the difference between the 25th and 75th percentile of a set of values and provides a trimmed estimation that is less sensitive to outliers compared to standard

deviation. This helps mitigate the impact of extreme values that may occur due to particularly favorable or unfavorable random initializations, providing a more reliable assessment of variation in MIA efficacy across different experimental repeats.

**Experimental Protocol**   Unless otherwise specified, all our results presented are averaged over 10 experimental repeats. Within each repeat, we sample a new subset of the population dataset (e.g. CIFAR-10). We sample the datasets to train the target and shadow models from the selected subset such that for each sample $x$ in the target model's training dataset, we have $1/2$ of the shadow models trained with $x$ whereas the remaining are not training with $x$. Additionally, we run the HPO algorithm for each experimental repeat to find optimal hyperparameters to train the target and shadow models. In order to ensure fair comparisons between different MIAs, we restrict our experiments to 1 target model per repeat as more target models are computationally expensive. With LiRA it is possible to use the so-called efficient LiRA implementation proposed by Carlini et al. (2022) that uses every shadow model also as a target model (see Section 5.3). While it is computationally cheap to implement the same also for RMIA, it is computationally expensive to do the same for the other attacks.

## 5   Results

We comprehensively analyze the factors that influence MIA efficacy of various attacks in transfer learning. In Section 5.1, we explore the effect of the number of shots on the performance of different attacks. In Section 5.2, we study how the performance of attacks varies for different parameterization schemes. In Section 5.3 and Section 5.4, we study the impact of number of shadow models and data augmentation on the most powerful MIAs, namely, LiRA and RMIA.

In addition, we investigate the impact of attack-specific parameters on MIA efficacy by evaluating how choice of attack threshold, $\gamma$, for RMIA (Appendix A.1) and distillation set size for Trajectory-MIA (Appendix A.2) affect their respective performance. These attacks have been proposed as more efficient alternatives to LiRA but we find their performance to be sensitive to the choice of hyperparameters, such as $\gamma$ in RMIA, introduced by the attack design.

### 5.1   Effect of Training Dataset Properties

Figure 2 demonstrates the relationships between the MIA efficacy and the number of examples per class, or shots ($S$), across all datasets, confirming the power law relationship proposed by Tobaben et al. (2024). Detailed numerical results are presented in Tables A2 to A4 and model performance is shown in Figure A3. Among all examined attacks, LiRA consistently outperforms other approaches across most experimental settings, exhibiting remarkable stability as evidenced by the narrower confidence intervals for it. The performance advantage of LiRA is particularly pronounced at lower $S$, though this advantage diminishes as $S$ increases.

While most attacks show monotonic degradation in MIA efficacy with increase in $S$, IHA displays non-monotonic patterns with multiple fluctuations. This could likely be due to the non-standard MIA threat model underlying IHA where the attacker has knowledge of all the remaining records (except the target sample) in the training dataset of the target model. This demonstrates that while increasing training data volume can effectively reduce average vulnerability to attacks in the standard MIA threat model (as shown by Tobaben et al. (2024)), the effect does not extend to attacks assuming a non-standard threat model, such as IHA.

### 5.2   Effect of Training Paradigm

Figure 3 illustrates the difference in MIA efficacy between Head-only, FiLM and ALL parameterizations for R-50 models fine-tuned using CIFAR-10 across multiple MIAs. Overall, we find that the strongest MIAs demonstrate minimal change in MIA efficacy due to change in parameterization. Given that LiRA maintains consistent performance across both parameterization schemes, this suggests that practitioners should choose parameterization scheme based primarily on utility considerations rather than privacy concerns, as the fine-

tuning choice does not significantly alter the efficacy of the most effective MIA approaches. Complete tabular data are provided in Table A5, and the model performance results are shown in Figure A4.

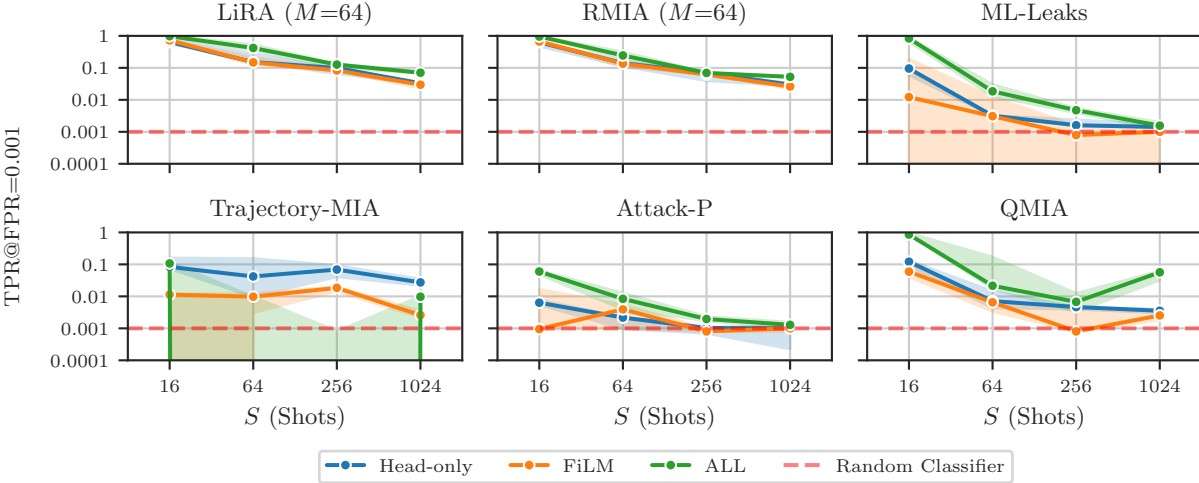

Figure 3: Comparison of MIA efficacy against R-50 fine-tuned on CIFAR-10 across 4 different $S$ (shots) with 3 different parameterization strategies: Head-only, FiLM, and ALL. The errorbars represent the interquartile range (IQR) of the estimated TPR at fixed FPR. Results are averaged over 5 repeats with 1 target model in each repeat. For the strongest attacks, there is no considerable difference in MIA efficacy across the 3 parameterization schemes. Some points for Trajectory-MIA with ALL fine-tuning are not visible in the plots due to poor performance or OOM issues.

## 5.3 Number of Shadow Models

Figure 4 illustrates the relationship between the number of shadow models $M$ and MIA efficacy for LiRA and RMIA, 2 of the strongest shadow-model-based attacks discussed in this paper. To ensure a statistically robust evaluation, we employ an efficient implementation of LiRA and RMIA proposed by Carlini et al. (2022). This approach involves sampling $M+1$ datasets from the training dataset $\mathcal{D}$, which contains $C \times S$ samples ($C$ classes with $S$ examples per class), such that each sample has a 0.5 probability of being selected for any given dataset. We then train models on each of these datasets and evaluate attacks against each model while using the remaining $M$ models as shadow models.

LiRA exhibits significant sensitivity to variations in $M$ across both low-data ($S = 16$) and data-rich ($S = 1024$) scenarios but is robust to increase in $M$ beyond $M \geq 64$. While RMIA demonstrates robustness to changes in $M$, its efficacy does not exceed LiRA in either of the scenarios. The performance for both the attacks stabilize beyond $M \geq 64$ suggesting it to be a cost-efficient choice for the number of shadow models in deep transfer learning setting. Full data are presented in Table A6.

## 5.4 Effect of Using Data Augmentation During Fine-Tuning

Prior research has demonstrated that both LiRA and RMIA benefit from querying each sample multiple times when attacking models trained from scratch using data augmentation (Carlini et al., 2022; Zarifzadeh et al., 2024). These attacks achieve improved performance by using not only the original sample but also augmented versions of it during the inference process. To determine whether similar improvements occur in deep transfer learning scenarios, we fine-tune target models using training datasets augmented with simple transformations, including mirror flipping and pixel shifting. This approach is employed in both from-scratch training (Perez & Wang, 2017) and transfer learning (Mehta et al., 2023) to improve model generalization, particularly when working with limited data.

Following the same efficient implementation as described in Section 5.3, we evaluated MIA efficacy using multiple augmented queries generated using a subset of transformations applied during training. For LiRA,

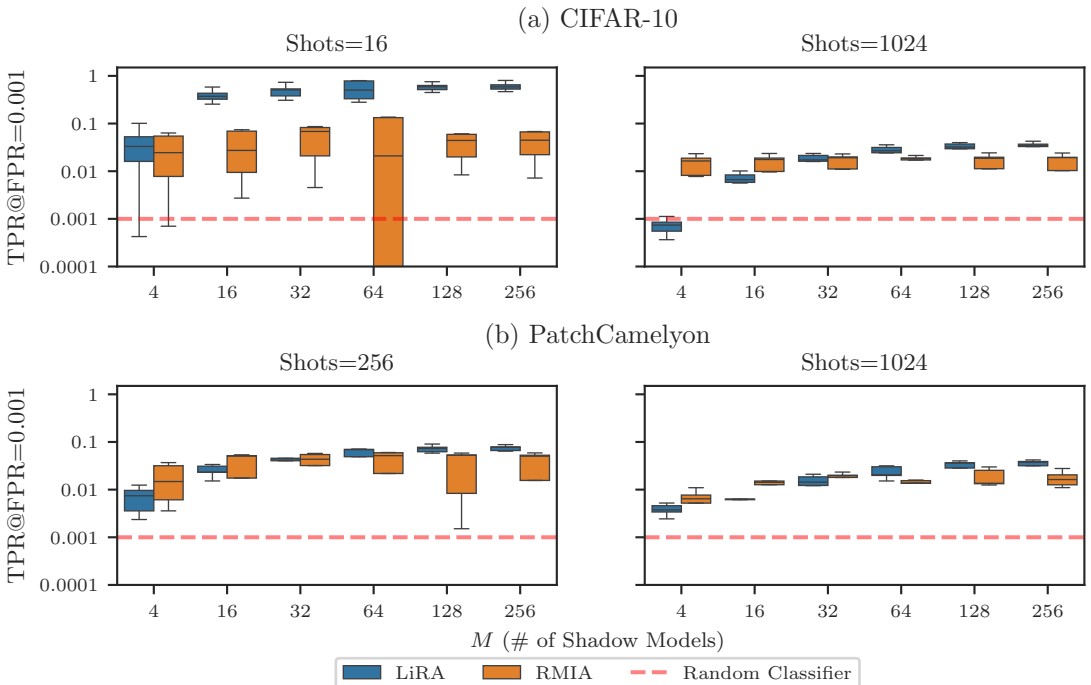

Figure 4: Relationship between MIA efficacy and the number of shadow models ($M$) for LiRA and RMIA against ViT-B/16 model with Head-only fine-tuned on (a) CIFAR-10 and (b) PatchCamelyon. Results demonstrates MIA efficacy in low data availability (shots $S = 16$ for CIFAR-10 and $S = 256$ for Patch-Camelyon) and high data availability ($S = 1024$ for both CIFAR-10 and PatchCamelyon) scenarios. For each configuration, we train $M + 1$ models per repeat, using each model as the target while the remaining $M$ serve as shadow models. We compute the average MIA efficacy (TPR at fixed FPR) across all $M + 1$ target models per repeat, then construct boxplots using these average TPR from 5 independent repeats. LiRA dominates in terms of efficacy over RMIA despite the latter's performance being more robust to the variations in $M$.

the membership signal is averaged over multiple queries directly, while for RMIA, we follow the majority voting scheme as recommended by Zarifzadeh et al. (2024). Figure 5 shows that data augmentation produces negligible performance improvements for both attacks when targeting Head-only fine-tuned models. This finding represents a significant departure from the from-scratch training findings, suggesting different vulnerability patterns in transfer learning. Furthermore, the test accuracy of the models trained with and without augmentations remain relatively close. Based on these results, we employ the non-augmented version of both attacks in all experiments for computational efficiency. Table A7 provides complete tabular data.

## 6 Discussion

Our findings largely corroborate the claim made by Tobaben et al. (2024) which states that increasing the number of examples per class generally reduces MIA efficacy. However, we do not find this behavior to be consistent across all the attacks. For example, MIA efficacy of IHA increases substantially at larger shots as observed in Figure 2. IHA does not use standard MIA threat model common to other attacks discussed in this paper. For a given target sample, it assumes knowledge of all $n-1$ records in a $n$-sized training dataset except for the target sample whereas in the standard MIA threat models, attacks rely on no such prior knowledge of other data records. The power law relationship between dataset size and the MIA performance proposed by Tobaben et al. (2024) holds for the standard MIA threat model and thus, does not extend to IHA.

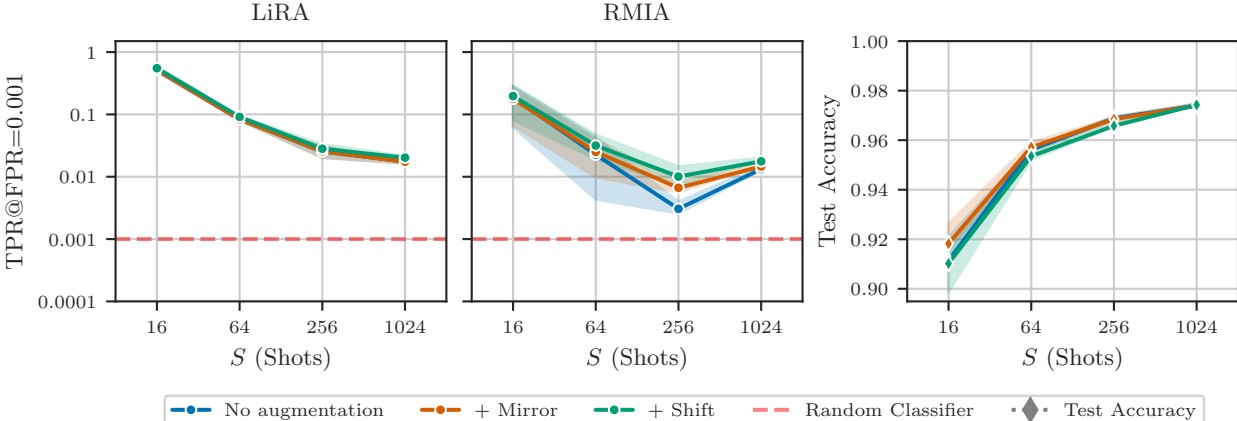

Figure 5: Impact of data augmentation on MIA efficacy across $S$ (shots) for ViT-B/16 models Head-only fine-tuned on CIFAR-10 *with data augmentations*. We compare 2 augmentation strategies: *+ Mirror* (where original image plus a horizontally flipped copy of it are used to train the target model) and *+ Shift* (where horizontally flipping and/or $\pm1$-pixel shifts are applied to the original image), with *No augmentation* as the baseline. The errorbars represent the interquartile range (IQR). Left and middle panels show MIA efficacy for LiRA and RMIA, respectively, while the right panel shows test accuracy. Results are averaged over 5 repeats and we use $M + 1$ target models ($M = 64$) per repeat.

For the most powerful attacks (e.g. LiRA) we find that different parameterization schemes, such as Head-only and FiLM, show minimal differences in terms of MIA efficacy. One notable exceptions is Trajectory-MIA, which shows increased vulnerability against Head-only fine-tuning. However, Trajectory-MIA is weaker and less stable compared to LiRA, which maintains consistent performance across both parameterization schemes. This implies that the parameterization choices for fine-tuning could be guided primarily by the utility as they do not significantly affect the performance of the strongest attacks.

Carlini et al. (2022); Zarifzadeh et al. (2024) suggest that data augmentation can be utilized to improve MIA efficacy against models trained from scratch. However, in Section 5.4 we observe no significant improvement in MIA efficacy due to augmentation against models with the last linear layer subject to fine-tuning.

No single MIA is able to capture all vulnerabilities in fine-tuned models. LiRA provides robust auditing capabilities but shows decreased efficacy as dataset sizes grow. IHA shows potential to detect vulnerabilities missed by black-box attacks, particularly with datasets that have different characteristics from the pre-training data, such as PatchCamelyon (Goyal et al., 2023; Choi et al., 2024; Thaker et al., 2024). Comprehensive privacy auditing requires a multi-faceted approach that combines both black-box and white-box methods.

**Limitations**

- We focus on balanced datasets in our experiments to evaluate the relationship between examples per class ($S$) and MIA efficacy for different attacks because this design choice enables clear comparison across different data availability scenarios. Future work could extend this analysis to imbalanced datasets commonly found in real-world deployments.

- To ensure a fair comparison across different attacks in sections 5.1 and 5.2, we restrict our experiments to having 1 target model per repeat. This differs from Carlini et al. (2022)'s efficient implementation of LiRA where they reuse all the $M + 1$ trained models as the target model and average the MIA efficacy over all of them. This is because attacks such as Trajectory-MIA require training 2 additional distilled models for each target model- shadow model pair, which makes the attack computationally infeasible if the number of target models per repeat is set to be the same

as LiRA ($M + 1$). Similar computational constraints are associated with IHA where approximating iHVPs per target sample per model will be prohibitively expensive for large numbers of target models.

- Another limitation of our paper is that we do not offer a detailed comparison between the performance of different MIAs against models trained with fine-tuning and from-scratch beyond Figure 1. This is because running these experiments for models trained from scratch would be computationally expensive.

# 7    Conclusion

In this work, we evaluated and compared the performance over a large set of MIAs in transfer learning setting. We found that the attack strength deteriorates as the dataset size increases for MIAs in the standard threat model. This agrees with the power law postulated by Tobaben et al. (2024). However, this relationship is not guaranteed to hold for all the attacks discussed in this paper, such as IHA operates under a threat model distinct from other attacks. This shows that there is no single existing attack that can fully quantify the privacy leakage in deep transfer learning. In addition, we found no significant difference in MIA efficacy between Head-only and FiLM parameterization strategies for most attacks, implying that choice of parameterization can be made with a utility-first perspective. However, MIAs are sensitive to the choice of attack properties, such as the number of shadow models used for the attack. These empirical findings provide guidance for practitioners seeking to assess privacy risks using MIAs in deep transfer learning applications.

The code for our experiments is available at:
`https://github.com/DPBayes/empirical-comparison-mia-transfer-learning`. We implement the attacks by adapting the code provided by prior works' authors [1] [2] [3] [4] [5] [6].

# Acknowledgments

This work was supported by the Research Council of Finland (Flagship programme: Finnish Center for Artificial Intelligence, FCAI, Grant 356499 and Grant 359111), the Strategic Research Council at the Research Council of Finland (Grant 358247) as well as the European Union (Project 101070617). Views and opinions expressed are however those of the author(s) only and do not necessarily reflect those of the European Union or the European Commission. Neither the European Union nor the granting authority can be held responsible for them. The authors wish to acknowledge CSC – IT Center for Science, Finland, for computational resources (Project 2003275).

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

## A.1 Influence of $\gamma$ in RMIA Scoring Function

As shown in Equation (2), RMIA uses $\gamma$ ($\geq 1$) as the threshold for the likelihood ratio test. It determines how much higher should the likelihood of observing model parameters $\theta$ be if a target sample $x$ was in the training dataset relative to a random population sample $z$ to pass the membership test. As such, $\gamma$ is a critical parameter in the RMIA. Following the same efficient implementation as described in Section 5.3, we conduct a sensitivity analysis varying $\gamma$ from 1 to 64 and report the MIA efficacy to evaluate attack performance. While RMIA efficacy against models trained from scratch is robust to the choice of $\gamma$ as shown by Zarifzadeh et al. (2024), this robustness does not extend to few-shot transfer learning setting. For ViT-B/16 models fine-tuned on CIFAR-10 with the Head-only setting, only RMIA with $\gamma = 2$ performs consistently across varying the number of shots. Despite wide error bars, its median performance remains stable.

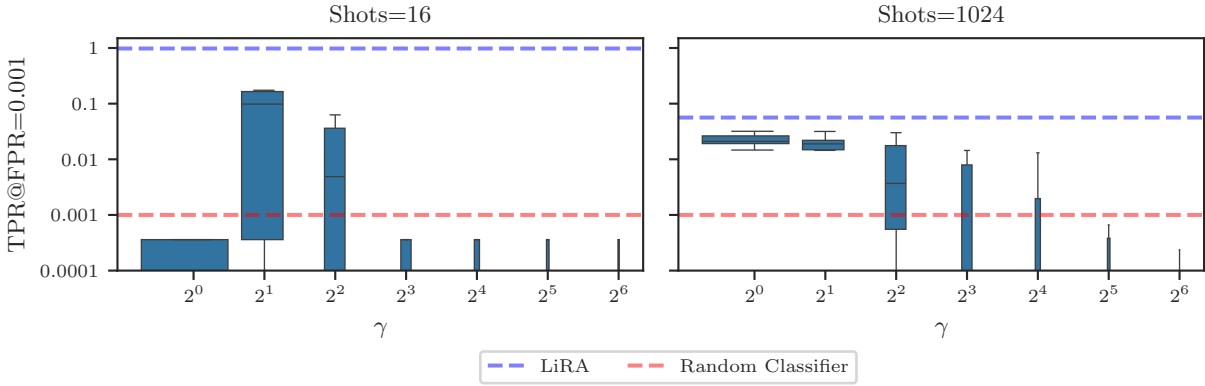

Figure A1: RMIA efficacy as a function of the threshold parameter $\gamma$. Results show MIA efficacy against ViT-B/16 models Head-only fine-tuned on CIFAR-10. We train $M + 1$ models ($M = 64$) per repeat, using each model as the target while the remaining $M$ serve as shadow models. We compute the average MIA efficacy (TPR at fixed FPR) across all $M + 1$ target models per repeat, then construct boxplots using the average TPR from 10 independent repeats. RMIA is shown to be sensitive to the value of $\gamma$ in deep transfer learning setting. The blue dashed line represents the median MIA efficacy achieved by LiRA under identical conditions.

## A.2 Effect of Distillation Set Size in Trajectory-MIA

Trajectory-MIA requires an auxiliary dataset for knowledge distillation to simulate the target model's training process. Following Liu et al. (2022) we split the given dataset into target, shadow, and distillation datasets, and we construct our distillation datasets using all data that is **not** a part of the target and shadow datasets.

Figure A2 illustrates that the distillation set size has to be sufficiently large for Trajectory-MIA to work effectively. Smaller distillation sets prevent the student model from learning sufficiently diverse examples to generalize to unseen data. However, $|\mathcal{D}^K| \geq 20000$ does not lead to any significant improvement in Trajectory-MIA efficacy. For CIFAR-10, using $|\mathcal{D}^K| = 20000$ for knowledge distillation proves to be more effective than using all available data ($\sim$50000 samples) in very low-shot regimes like $S = 16$. As the number of shots increases, the performance difference between $|\mathcal{D}^K| = 20000$ and $|\mathcal{D}^K| = $ All decreases, because the available data for distillation become increasingly similar in both scenarios.

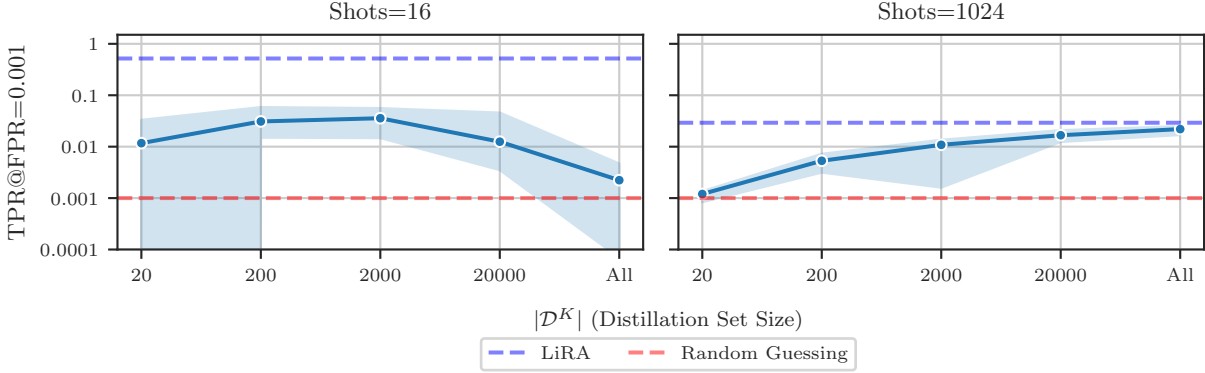

Figure A2: Sensitivity of Trajectory-MIA efficacy to distillation set size. MIA efficacy is evaluated against ViT-B/16 models Head-only fine-tuned on CIFAR-10. $|\mathcal{D}^K|$ represents the distillation set size, with All representing using all available data not part of the fine-tuning datasets. The errorbars represent the interquartile range (IQR) associated with the estimated TPR at fixed FPR. The blue dashed line represents the median MIA efficacy achieved by LiRA under identical conditions. We use LiRA with 64 shadow models as the upper bound on MIA efficacy. Results are averaged over 10 repeats and we use 1 target model per repeat.

## A.3 Additional Results

### A.3.1 Model Performance

The training and test accuracies for the target models are plotted in Figure A3 for the ViT-B/16 models used in Sections 5.1 and 5.3, and in Figure A4 for the R-50 models used in Section 5.2. Figure A3 indicates that datasets exhibiting a large distribution shift from the pretraining data (such as PatchCamelyon) are associated with lower training and test accuracies compared to datasets that closely match the pre-training distribution (such as CIFAR-10). This is consistent with degraded feature transferability under distribution shift. Figure A4 shows that Head-only, FiLM, and ALL fine-tuning achieve comparable difference in model performance on the training and test sets, which explains why the choice of training paradigm has little effect on MIA efficacy for the strongest attacks.

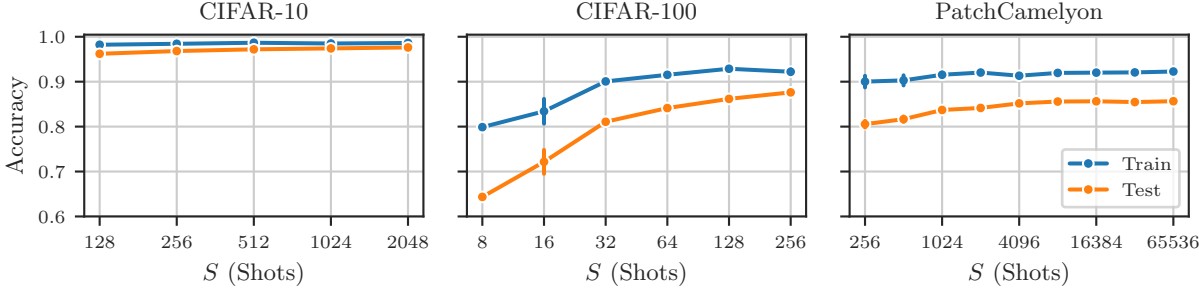

Figure A3: Training and test accuracies for ViT-B/16 models pre-trained on ImageNet-21k and Head-only fine-tuned with varying $S$ (shots). The errorbars represent the standard errors. Results are averaged over 10 repeats and we use accuracies from 10 random models per repeat.

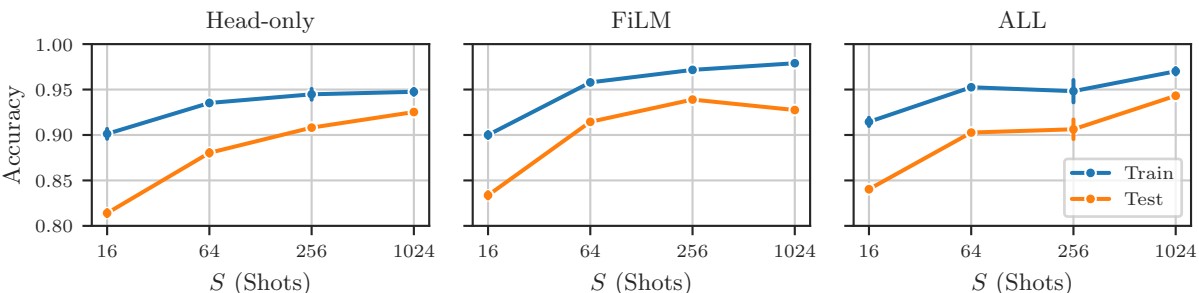

Figure A4: Training and test accuracies for R-50 models pre-trained on ImageNet-21k fine-tuned on CIFAR-10 across varying $S$ (shots) and 3 different parameterization strategies: *Head-only*, *FiLM*, and *ALL*. The errorbars represent the standard errors. Results are averaged over 10 repeats and we use accuracies from 10 random models per repeat. The gap between training and test accuracy decreases as $S$ increases, indicating reduced overfitting. This aligns with the observed decline in MIA efficacy, as overfitting is strongly correlated with attack efficacy.

### A.3.2 ROC Plots Comparing MIA Efficacy Across Different Attacks

Figures A5 to A7 represent full ROC curves for attacks used in Figure 2.

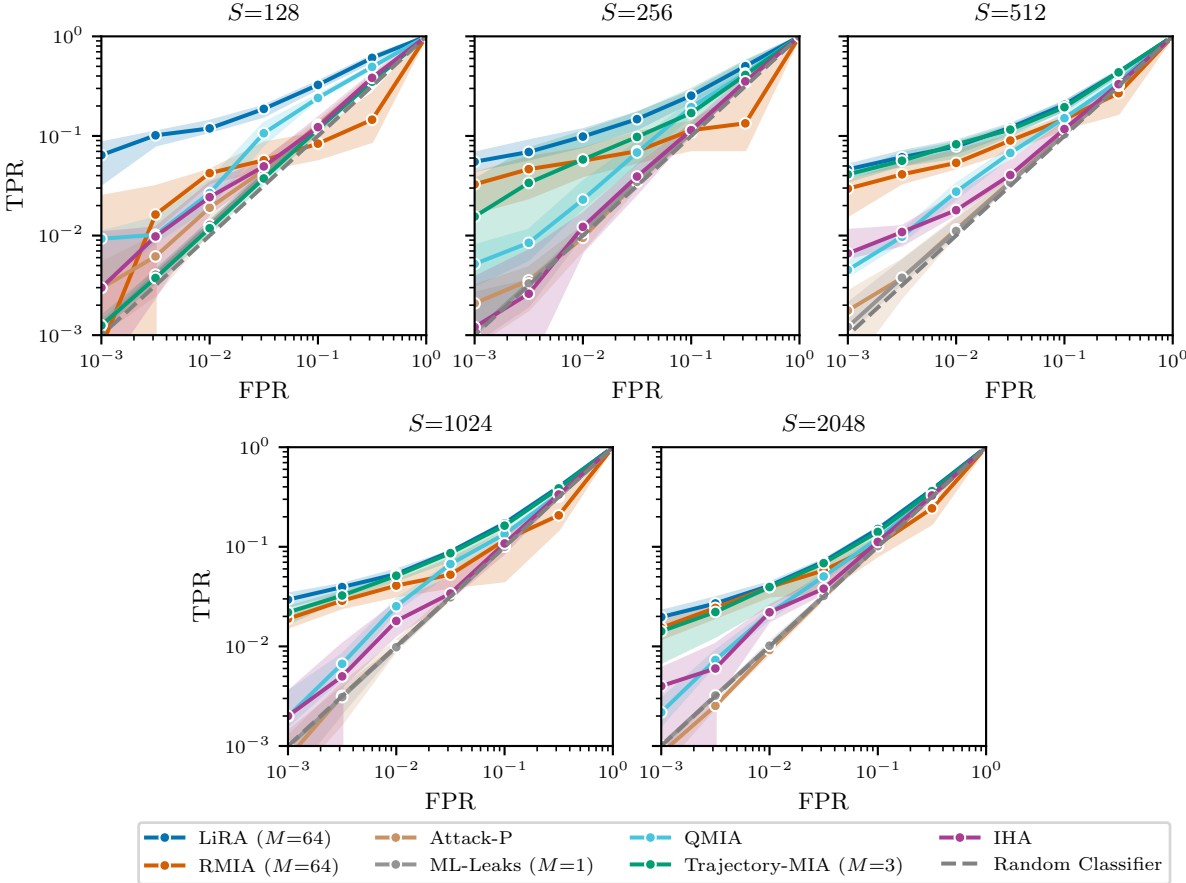

Figure A5: ROC curves for ViT-B/16 models Head-only fine-tuned on CIFAR-10 with varying $S$ (shots). The errorbars represent the interquartile range (IQR) associated with the estimated TPR at fixed FPR. Results are averaged over 10 repeats and we use 1 target model per repeat.

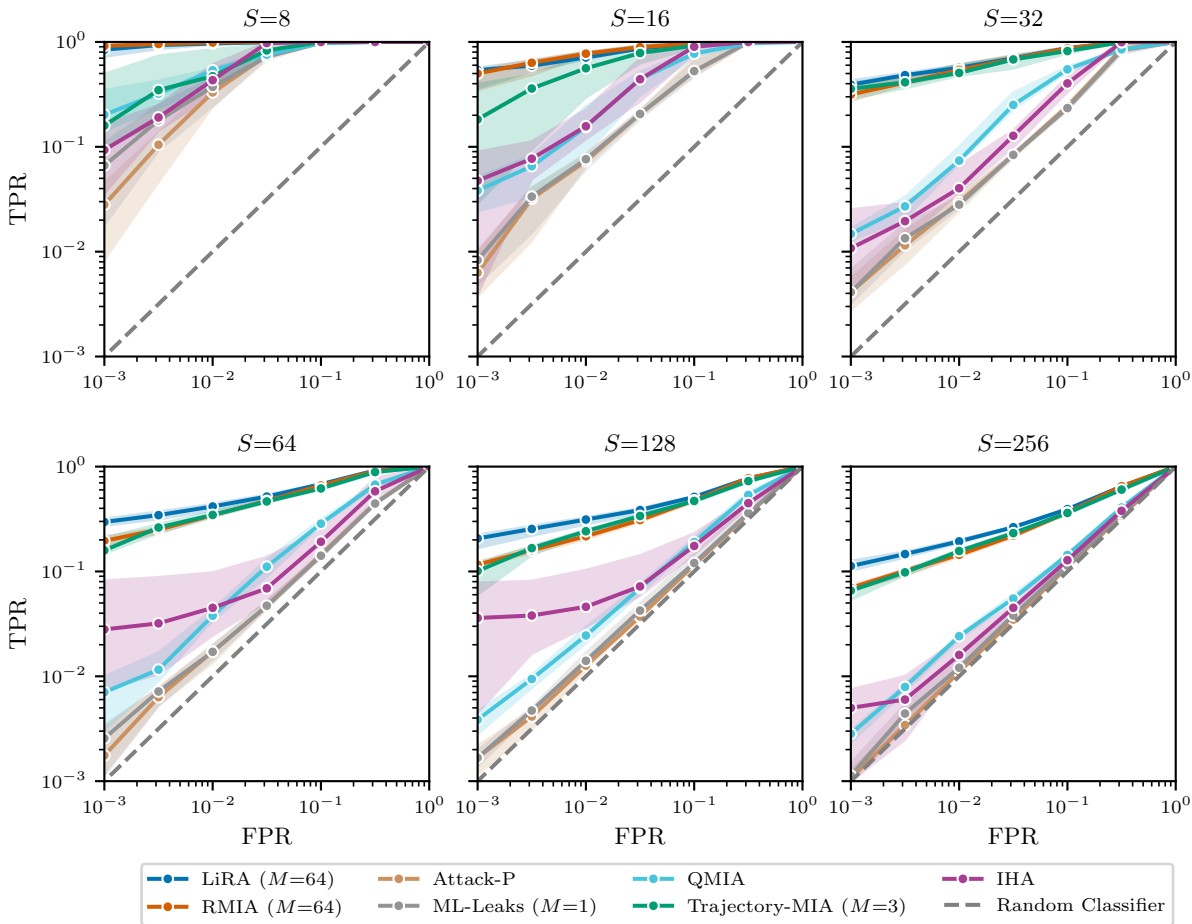

Figure A6: ROC curves for ViT-B/16 models Head-only fine-tuned on CIFAR-100 with varying $S$ (shots). The errorbars represent the interquartile range (IQR) associated with the estimated TPR at fixed FPR. Results are averaged over 10 repeats and we use 1 target model per repeat.

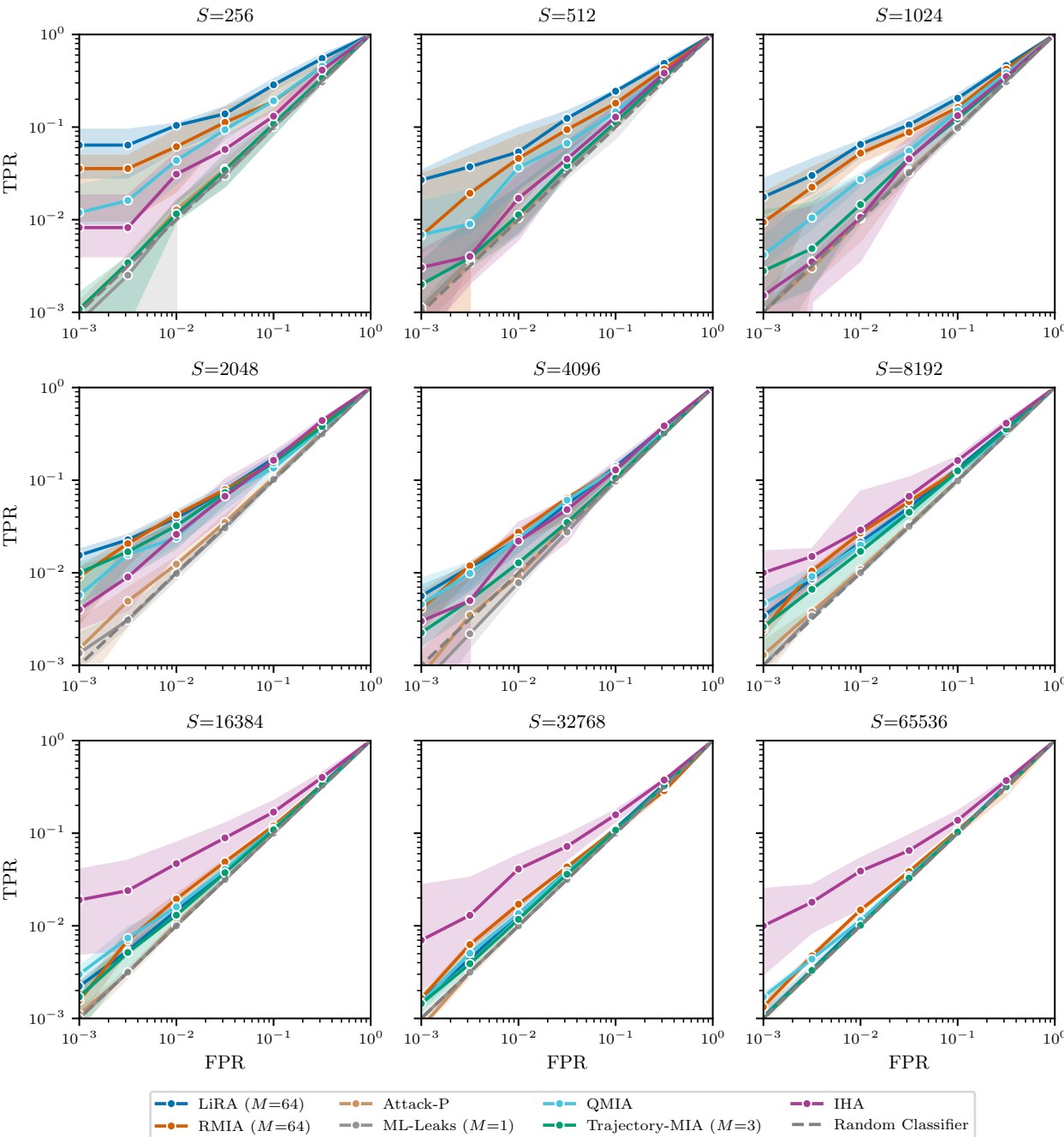

Figure A7: ROC curves for ViT-B/16 models Head-only fine-tuned on PatchCamelyon with varying $S$ (shots). The errorbars represent the interquartile range (IQR) associated with the estimated TPR at fixed FPR. Results are averaged over 10 repeats and we use 1 target model per repeat.

### A.3.3 Effect of Domain Shift on MIA Efficacy

To assess MIA efficacy when target models are fine-tuned on datasets that are out-of-distribution (OOD) relative to the pre-training data, we perform the same analysis as in Section 5.2 using PatchCamelyon. Results are presented in Appendix A.3.3 and Table A1. Consistent with the findings in Section 5.2, we observe that the strongest MIAs show minimal sensitivity to parameterization changes. Note that Trajectory-MIA results are omitted due to CUDA memory limitations while training the distilled versions of shadow and target model required for this attack.

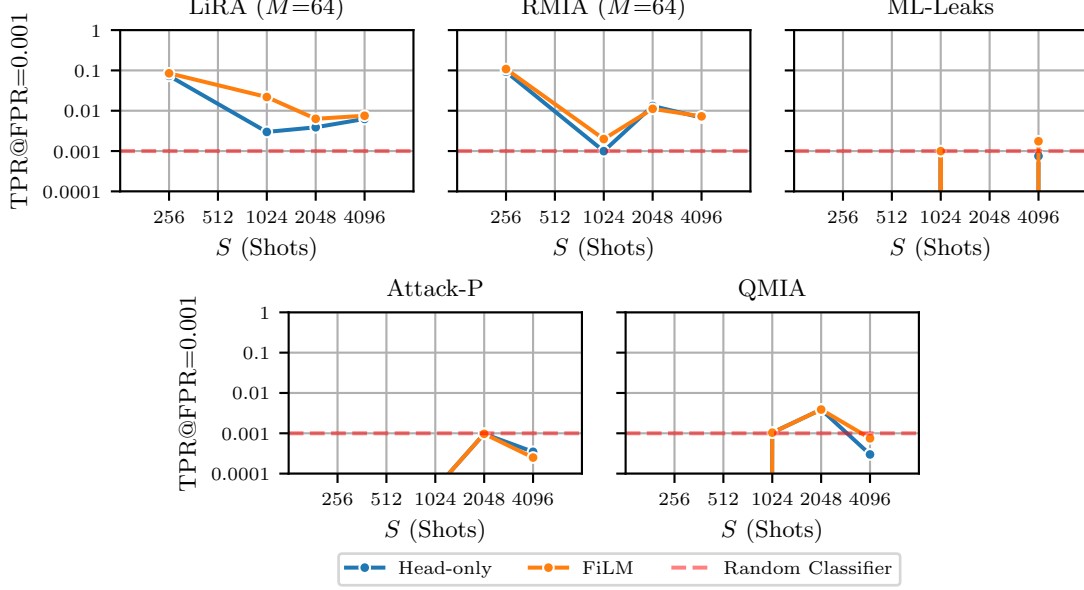

Figure A8: Comparison of MIA efficacy against R-50 fine-tuned on PatchCamelyon with Head-only versus FiLM parameterization across 4 different $S$ (shots). Some points are not visible in the plots due to poor performance (TPR< 0.0001).

Table A1: MIA efficacy against R-50 models fine-tuned on PatchCamelyon with Head-only versus FiLM parameterization across 4 different $S$ (shots).

| Shots | Attack | Parameterization | TPR@FPR=0.1 | TPR@FPR=0.01 | TPR@FPR=0.001 |
|---|---|---|---|---|---|
| $S = 256$ | LiRA ($M = 64$) | Head-only | **0.340** | **0.135** | 0.073 |
| | | FiLM | **0.332** | **0.135** | 0.085 |
| | RMIA ($M = 64$) | Head-only | 0.093 | 0.093 | **0.093** |
| | | FiLM | 0.108 | 0.108 | **0.108** |
| | ML-Leaks ($M = 1$) | Head-only | 0.093 | 0.000 | 0.000 |
| | | FiLM | 0.093 | 0.000 | 0.000 |
| | Attack-P | Head-only | 0.085 | 0.000 | 0.000 |
| | | FiLM | 0.087 | 0.000 | 0.000 |
| | QMIA | Head-only | 0.132 | 0.000 | 0.000 |
| | | FiLM | 0.137 | 0.000 | 0.000 |
| | LiRA ($M = 64$) | Head-only | **0.158** | **0.030** | **0.003** |
| | | FiLM | **0.213** | **0.049** | 0.022 |
| | RMIA ($M = 64$) | Head-only | 0.128 | **0.032** | **0.001** |
| | | FiLM | 0.174 | 0.031 | 0.002 |

**Table A1 – continued from previous page**

| Shots | Attack | Parameterization | TPR@FPR=0.1 | TPR@FPR=0.01 | TPR@FPR=0.001 |
|---|---|---|---|---|---|
| $S = 1024$ | ML-Leaks ($M = 1$) | Head-only | 0.114 | 0.005 | 0.001 |
| | | FiLM | 0.112 | 0.007 | 0.001 |
| | Attack-P | Head-only | 0.115 | 0.005 | 0.000 |
| | | FiLM | 0.113 | 0.006 | 0.000 |
| | QMIA | Head-only | 0.122 | 0.009 | 0.001 |
| | | FiLM | 0.131 | 0.010 | 0.001 |
| $S = 2048$ | LiRA ($M = 64$) | Head-only | **0.178** | **0.046** | 0.004 |
| | | FiLM | **0.178** | 0.035 | 0.006 |
| | RMIA ($M = 64$) | Head-only | 0.116 | 0.029 | **0.013** |
| | | FiLM | 0.157 | **0.046** | **0.011** |
| | ML-Leaks ($M = 1$) | Head-only | 0.100 | 0.008 | 0.000 |
| | | FiLM | 0.097 | 0.008 | 0.000 |
| | Attack-P | Head-only | 0.094 | 0.013 | 0.001 |
| | | FiLM | 0.100 | 0.010 | 0.001 |
| | QMIA | Head-only | 0.126 | 0.015 | 0.004 |
| | | FiLM | 0.118 | 0.015 | 0.004 |
| $S = 4096$ | LiRA ($M = 64$) | Head-only | **0.148** | 0.028 | **0.006** |
| | | FiLM | 0.169 | 0.029 | **0.008** |
| | RMIA ($M = 64$) | Head-only | 0.128 | **0.042** | **0.007** |
| | | FiLM | **0.186** | 0.034 | **0.007** |
| | ML-Leaks ($M = 1$) | Head-only | 0.107 | 0.013 | 0.001 |
| | | FiLM | 0.101 | 0.009 | 0.002 |
| | Attack-P | Head-only | 0.107 | 0.013 | 0.000 |
| | | FiLM | 0.099 | 0.012 | 0.000 |
| | QMIA | Head-only | 0.096 | 0.011 | 0.000 |
| | | FiLM | 0.099 | 0.010 | 0.001 |

### A.3.4 Additional Effect of Training Dataset Properties

Tables A2 to A4 provide tabular results corresponding to Figure 2, demonstrating how MIA efficacy varies as a function of $S$ (shots) across three datasets: CIFAR-10, CIFAR-100, and PatchCamelyon. The tables present numerical data for both shadow-model-based attacks (using varying numbers of shadow models $M$) and shadow-model-free attacks, with results obtained using ViT-B/16 models pre-trained on ImageNet-21k and fine-tuned with Head-only setting.

Table A2: MIA efficacy as a function of $S$ (shots) for ViT-B/16 models pre-trained on ImageNet-21k and fine-tuned on CIFAR-10 (Head-only). Values represent medians with interquartile ranges (IQRs). Results are averaged over 10 repeats and we use 1 target model per repeat.

| Shots | Attack | TPR@FPR=0.1 | | TPR@FPR=0.01 | | TPR@FPR=0.001 | |
|---|---|---|---|---|---|---|---|
| | | Median | IQR | Median | IQR | Median | IQR |
| $S = 128$ | LiRA ($M = 64$) | **0.33** | 0.05 | **0.12** | 0.03 | **0.06** | 0.05 |
| | RMIA ($M = 64$) | 0.08 | 0.05 | 0.04 | 0.01 | 0.00 | 0.02 |
| | ML-Leaks ($M = 1$) | 0.13 | 0.02 | 0.01 | 0.00 | 0.00 | 0.00 |
| | Trajectory-MIA ($M = 3$) | 0.12 | 0.02 | 0.01 | 0.00 | 0.00 | 0.00 |
| | Attack-P | 0.13 | 0.02 | 0.02 | 0.01 | 0.00 | 0.00 |
| | QMIA | 0.24 | 0.02 | 0.03 | 0.02 | 0.01 | 0.01 |
| | IHA | 0.12 | 0.04 | 0.02 | 0.02 | 0.00 | 0.01 |
| $S = 256$ | LiRA ($M = 64$) | **0.25** | 0.06 | **0.10** | 0.04 | **0.06** | 0.02 |
| | RMIA ($M = 64$) | 0.11 | 0.07 | **0.06** | 0.04 | 0.03 | 0.03 |
| | ML-Leaks ($M = 1$) | 0.11 | 0.01 | 0.01 | 0.00 | 0.00 | 0.00 |
| | Trajectory-MIA ($M = 3$) | 0.17 | 0.18 | **0.06** | 0.10 | 0.02 | 0.04 |
| | Attack-P | 0.11 | 0.01 | 0.01 | 0.01 | 0.00 | 0.00 |
| | QMIA | **0.19** | 0.06 | 0.02 | 0.02 | 0.01 | 0.00 |
| | IHA | 0.11 | 0.02 | 0.01 | 0.01 | 0.00 | 0.00 |
| $S = 512$ | LiRA ($M = 64$) | **0.21** | 0.04 | **0.08** | 0.02 | **0.05** | 0.01 |
| | RMIA ($M = 64$) | 0.15 | 0.04 | 0.05 | 0.02 | 0.03 | 0.02 |
| | ML-Leaks ($M = 1$) | 0.10 | 0.01 | 0.01 | 0.00 | 0.00 | 0.00 |
| | Trajectory-MIA ($M = 3$) | **0.19** | 0.04 | **0.08** | 0.02 | **0.04** | 0.01 |
| | Attack-P | 0.11 | 0.01 | 0.01 | 0.00 | 0.00 | 0.00 |
| | QMIA | 0.15 | 0.03 | 0.03 | 0.01 | 0.00 | 0.00 |
| | IHA | 0.12 | 0.03 | 0.02 | 0.01 | 0.01 | 0.01 |
| $S = 1024$ | LiRA ($M = 64$) | **0.17** | 0.01 | **0.05** | 0.01 | **0.03** | 0.01 |
| | RMIA ($M = 64$) | 0.12 | 0.10 | **0.04** | 0.01 | **0.02** | 0.01 |
| | ML-Leaks ($M = 1$) | 0.10 | 0.01 | 0.01 | 0.00 | 0.00 | 0.00 |
| | Trajectory-MIA ($M = 3$) | **0.16** | 0.01 | **0.05** | 0.01 | **0.02** | 0.01 |
| | Attack-P | 0.10 | 0.00 | 0.01 | 0.00 | 0.00 | 0.00 |
| | QMIA | 0.13 | 0.02 | 0.03 | 0.01 | 0.00 | 0.00 |
| | IHA | 0.11 | 0.03 | 0.02 | 0.01 | 0.00 | 0.00 |
| $S = 2048$ | LiRA ($M = 64$) | **0.15** | 0.01 | **0.04** | 0.01 | **0.02** | 0.01 |
| | RMIA ($M = 64$) | 0.11 | 0.04 | **0.04** | 0.01 | **0.02** | 0.01 |
| | ML-Leaks ($M = 1$) | 0.10 | 0.00 | 0.01 | 0.00 | 0.00 | 0.00 |
| | Trajectory-MIA ($M = 3$) | **0.14** | 0.02 | **0.04** | 0.01 | 0.01 | 0.01 |
| | Attack-P | 0.10 | 0.01 | 0.01 | 0.00 | 0.00 | 0.00 |
| | QMIA | 0.12 | 0.01 | 0.02 | 0.00 | 0.00 | 0.00 |
| | IHA | 0.12 | 0.01 | 0.02 | 0.01 | 0.00 | 0.01 |

Table A3: MIA efficacy as a function of $S$ (shots) for ViT-B/16 models pre-trained on ImageNet-21k and fine-tuned on CIFAR-100 (Head-only). Values represent medians with interquartile ranges (IQRs). Results are averaged over 10 repeats and we use 1 target model per repeat.

| Shots | Attack | TPR@FPR=0.1 | | TPR@FPR=0.01 | | TPR@FPR=0.001 | |
|---|---|---|---|---|---|---|---|
| | | Median | IQR | Median | IQR | Median | IQR |
| $S = 8$ | LiRA ($M = 64$) | **1.00** | 0.00 | **0.97** | 0.03 | 0.84 | 0.20 |
| | RMIA ($M = 64$) | **1.00** | 0.00 | **0.99** | 0.03 | **0.92** | 0.07 |
| | ML-Leaks ($M = 1$) | 0.98 | 0.02 | 0.37 | 0.22 | 0.07 | 0.09 |
| | Trajectory-MIA ($M = 3$) | **1.00** | 0.01 | 0.47 | 0.46 | 0.16 | 0.40 |
| | Attack-P | 0.99 | 0.02 | 0.33 | 0.17 | 0.03 | 0.04 |
| | QMIA | 0.98 | 0.02 | 0.54 | 0.16 | 0.20 | 0.20 |
| | IHA | **1.00** | 0.00 | 0.43 | 0.30 | 0.09 | 0.13 |
| $S = 16$ | LiRA ($M = 64$) | **0.97** | 0.04 | **0.71** | 0.12 | **0.54** | 0.21 |
| | RMIA ($M = 64$) | **0.97** | 0.04 | **0.77** | 0.06 | **0.50** | 0.19 |
| | ML-Leaks ($M = 1$) | 0.53 | 0.19 | 0.08 | 0.02 | 0.01 | 0.01 |
| | Trajectory-MIA ($M = 3$) | 0.92 | 0.11 | 0.56 | 0.46 | 0.18 | 0.37 |
| | Attack-P | 0.52 | 0.23 | 0.07 | 0.02 | 0.01 | 0.01 |
| | QMIA | 0.77 | 0.13 | 0.15 | 0.16 | 0.04 | 0.03 |
| | IHA | 0.90 | 0.22 | 0.16 | 0.11 | 0.05 | 0.09 |
| $S = 32$ | LiRA ($M = 64$) | **0.87** | 0.04 | **0.57** | 0.04 | **0.39** | 0.10 |
| | RMIA ($M = 64$) | **0.87** | 0.03 | **0.54** | 0.10 | **0.32** | 0.09 |
| | ML-Leaks ($M = 1$) | 0.23 | 0.03 | 0.03 | 0.01 | 0.00 | 0.00 |
| | Trajectory-MIA ($M = 3$) | 0.82 | 0.10 | 0.51 | 0.13 | **0.36** | 0.12 |
| | Attack-P | 0.24 | 0.03 | 0.03 | 0.01 | 0.00 | 0.00 |
| | QMIA | 0.55 | 0.07 | 0.07 | 0.03 | 0.01 | 0.00 |
| | IHA | 0.40 | 0.13 | 0.04 | 0.03 | 0.01 | 0.02 |
| $S = 64$ | LiRA ($M = 64$) | **0.68** | 0.03 | **0.42** | 0.06 | **0.30** | 0.04 |
| | RMIA ($M = 64$) | **0.66** | 0.03 | 0.34 | 0.02 | 0.20 | 0.04 |
| | ML-Leaks ($M = 1$) | 0.14 | 0.01 | 0.02 | 0.01 | 0.00 | 0.00 |
| | Trajectory-MIA ($M = 3$) | 0.62 | 0.05 | 0.35 | 0.04 | 0.16 | 0.06 |
| | Attack-P | 0.14 | 0.01 | 0.02 | 0.01 | 0.00 | 0.00 |
| | QMIA | 0.29 | 0.04 | 0.04 | 0.01 | 0.01 | 0.01 |
| | IHA | 0.20 | 0.09 | 0.04 | 0.07 | 0.03 | 0.07 |
| $S = 128$ | LiRA ($M = 64$) | **0.51** | 0.03 | **0.31** | 0.05 | **0.21** | 0.06 |
| | RMIA ($M = 64$) | **0.48** | 0.03 | 0.22 | 0.02 | 0.12 | 0.01 |
| | ML-Leaks ($M = 1$) | 0.12 | 0.03 | 0.01 | 0.00 | 0.00 | 0.00 |
| | Trajectory-MIA ($M = 3$) | 0.47 | 0.02 | 0.24 | 0.03 | 0.10 | 0.06 |
| | Attack-P | 0.12 | 0.01 | 0.01 | 0.00 | 0.00 | 0.00 |
| | QMIA | 0.19 | 0.03 | 0.02 | 0.01 | 0.00 | 0.00 |
| | IHA | 0.18 | 0.09 | 0.05 | 0.07 | 0.04 | 0.07 |
| $S = 256$ | LiRA ($M = 64$) | **0.39** | 0.01 | **0.19** | 0.01 | **0.11** | 0.02 |
| | RMIA ($M = 64$) | 0.36 | 0.02 | 0.14 | 0.01 | 0.07 | 0.01 |
| | ML-Leaks ($M = 1$) | 0.11 | 0.02 | 0.01 | 0.00 | 0.00 | 0.00 |
| | Trajectory-MIA ($M = 3$) | 0.36 | 0.01 | 0.16 | 0.02 | 0.07 | 0.02 |
| | Attack-P | 0.11 | 0.01 | 0.01 | 0.00 | 0.00 | 0.00 |
| | QMIA | 0.14 | 0.01 | 0.02 | 0.00 | 0.00 | 0.00 |
| | IHA | 0.14 | 0.04 | 0.02 | 0.01 | 0.01 | 0.01 |

Table A4: MIA efficacy as a function of $S$ (shots) for ViT-B/16 models pre-trained on ImageNet-21k and fine-tuned on PatchCamelyon (Head-only). Values represent medians with interquartile ranges (IQRs). Results are averaged over 10 repeats and we use 1 target model per repeat.

| Shots | Attack | TPR@FPR=0.1 | | TPR@FPR=0.01 | | TPR@FPR=0.001 | |
|---|---|---|---|---|---|---|---|
| | | Median | IQR | Median | IQR | Median | IQR |
| $S = 256$ | LiRA ($M = 64$) | **0.29** | 0.08 | **0.10** | 0.05 | **0.06** | 0.07 |
| | RMIA ($M = 64$) | 0.19 | 0.12 | **0.06** | 0.07 | **0.04** | 0.04 |
| | ML-Leaks ($M = 1$) | 0.10 | 0.02 | 0.01 | 0.01 | 0.00 | 0.00 |
| | Trajectory-MIA ($M = 3$) | 0.11 | 0.03 | 0.01 | 0.01 | 0.00 | 0.00 |
| | Attack-P | 0.10 | 0.02 | 0.01 | 0.01 | 0.00 | 0.00 |
| | QMIA | 0.19 | 0.06 | 0.04 | 0.03 | 0.01 | 0.01 |
| | IHA | 0.13 | 0.03 | 0.03 | 0.04 | 0.01 | 0.01 |
| $S = 512$ | LiRA ($M = 64$) | **0.24** | 0.10 | **0.05** | 0.08 | **0.03** | 0.03 |
| | RMIA ($M = 64$) | **0.18** | 0.09 | **0.05** | 0.07 | **0.01** | 0.03 |
| | ML-Leaks ($M = 1$) | 0.11 | 0.04 | 0.01 | 0.00 | 0.00 | 0.00 |
| | Trajectory-MIA ($M = 3$) | 0.11 | 0.03 | 0.01 | 0.02 | 0.00 | 0.00 |
| | Attack-P | 0.11 | 0.01 | 0.01 | 0.00 | 0.00 | 0.00 |
| | QMIA | 0.15 | 0.05 | **0.04** | 0.03 | **0.01** | 0.01 |
| | IHA | 0.13 | 0.06 | 0.02 | 0.02 | 0.00 | 0.00 |
| $S = 1024$ | LiRA ($M = 64$) | **0.20** | 0.04 | **0.07** | 0.02 | **0.02** | 0.02 |
| | RMIA ($M = 64$) | **0.16** | 0.06 | **0.05** | 0.02 | **0.01** | 0.02 |
| | ML-Leaks ($M = 1$) | 0.10 | 0.01 | 0.01 | 0.01 | 0.00 | 0.00 |
| | Trajectory-MIA ($M = 3$) | 0.12 | 0.07 | 0.01 | 0.02 | 0.00 | 0.01 |
| | Attack-P | 0.10 | 0.01 | 0.01 | 0.01 | 0.00 | 0.00 |
| | QMIA | 0.15 | 0.04 | 0.03 | 0.01 | 0.00 | 0.00 |
| | IHA | 0.13 | 0.05 | 0.01 | 0.02 | 0.00 | 0.01 |
| $S = 2048$ | LiRA ($M = 64$) | **0.17** | 0.02 | **0.04** | 0.01 | **0.02** | 0.01 |
| | RMIA ($M = 64$) | **0.16** | 0.02 | **0.04** | 0.01 | **0.01** | 0.01 |
| | ML-Leaks ($M = 1$) | 0.10 | 0.01 | 0.01 | 0.00 | 0.00 | 0.00 |
| | Trajectory-MIA ($M = 3$) | **0.15** | 0.04 | **0.03** | 0.02 | **0.01** | 0.01 |
| | Attack-P | 0.11 | 0.01 | 0.01 | 0.00 | 0.00 | 0.00 |
| | QMIA | 0.14 | 0.04 | 0.02 | 0.02 | **0.01** | 0.01 |
| | IHA | **0.16** | 0.05 | **0.03** | 0.02 | 0.00 | 0.00 |
| $S = 4096$ | LiRA ($M = 64$) | **0.14** | 0.03 | **0.02** | 0.01 | **0.01** | 0.00 |
| | RMIA ($M = 64$) | **0.13** | 0.02 | **0.03** | 0.01 | 0.00 | 0.00 |
| | ML-Leaks ($M = 1$) | 0.10 | 0.01 | 0.01 | 0.00 | 0.00 | 0.00 |
| | Trajectory-MIA ($M = 3$) | 0.11 | 0.04 | 0.01 | 0.01 | 0.00 | 0.00 |
| | Attack-P | 0.10 | 0.01 | 0.01 | 0.00 | 0.00 | 0.00 |
| | QMIA | **0.13** | 0.02 | **0.02** | 0.00 | 0.00 | 0.00 |
| | IHA | **0.13** | 0.04 | **0.02** | 0.03 | 0.00 | 0.00 |
| $S = 8192$ | LiRA ($M = 64$) | 0.13 | 0.01 | 0.02 | 0.00 | 0.00 | 0.00 |
| | RMIA ($M = 64$) | 0.13 | 0.01 | **0.03** | 0.00 | 0.00 | 0.00 |
| | ML-Leaks ($M = 1$) | 0.10 | 0.01 | 0.01 | 0.00 | 0.00 | 0.00 |
| | Trajectory-MIA ($M = 3$) | 0.13 | 0.03 | 0.02 | 0.01 | 0.00 | 0.00 |
| | Attack-P | 0.10 | 0.01 | 0.01 | 0.00 | 0.00 | 0.00 |
| | QMIA | 0.12 | 0.01 | 0.02 | 0.01 | 0.00 | 0.00 |
| | IHA | **0.17** | 0.03 | **0.03** | 0.05 | **0.01** | 0.01 |
| | LiRA ($M = 64$) | 0.11 | 0.01 | 0.01 | 0.00 | 0.00 | 0.00 |
| | RMIA ($M = 64$) | **0.12** | 0.01 | **0.02** | 0.01 | 0.00 | 0.00 |

**Table A4 – continued from previous page**

| Shots | Attack | TPR@FPR=0.1 | | TPR@FPR=0.01 | | TPR@FPR=0.001 | |
|---|---|---|---|---|---|---|---|
| | | Median | IQR | Median | IQR | Median | IQR |
| $S = 16384$ | ML-Leaks ($M = 1$) | 0.10 | 0.00 | 0.01 | 0.00 | 0.00 | 0.00 |
| | Trajectory-MIA ($M = 3$) | 0.11 | 0.01 | 0.01 | 0.00 | 0.00 | 0.00 |
| | Attack-P | 0.10 | 0.00 | 0.01 | 0.00 | 0.00 | 0.00 |
| | QMIA | 0.11 | 0.01 | **0.02** | 0.00 | 0.00 | 0.00 |
| | IHA | **0.17** | 0.09 | **0.05** | 0.06 | **0.02** | 0.04 |
| $S = 32768$ | LiRA ($M = 64$) | 0.11 | 0.01 | 0.01 | 0.00 | 0.00 | 0.00 |
| | RMIA ($M = 64$) | 0.11 | 0.02 | **0.02** | 0.00 | 0.00 | 0.00 |
| | ML-Leaks ($M = 1$) | 0.10 | 0.00 | 0.01 | 0.00 | 0.00 | 0.00 |
| | Trajectory-MIA ($M = 3$) | 0.11 | 0.00 | 0.01 | 0.00 | 0.00 | 0.00 |
| | Attack-P | 0.10 | 0.00 | 0.01 | 0.00 | 0.00 | 0.00 |
| | QMIA | 0.11 | 0.01 | 0.01 | 0.00 | 0.00 | 0.00 |
| | IHA | **0.16** | 0.04 | **0.04** | 0.04 | **0.01** | 0.03 |
| $S = 65536$ | LiRA ($M = 64$) | 0.10 | 0.01 | 0.01 | 0.00 | 0.00 | 0.00 |
| | RMIA ($M = 64$) | **0.11** | 0.01 | 0.01 | 0.00 | 0.00 | 0.00 |
| | ML-Leaks ($M = 1$) | 0.10 | 0.00 | 0.01 | 0.00 | 0.00 | 0.00 |
| | Trajectory-MIA ($M = 3$) | 0.10 | 0.00 | 0.01 | 0.00 | 0.00 | 0.00 |
| | Attack-P | 0.10 | 0.00 | 0.01 | 0.00 | 0.00 | 0.00 |
| | QMIA | 0.10 | 0.00 | 0.01 | 0.00 | 0.00 | 0.00 |
| | IHA | **0.14** | 0.05 | **0.04** | 0.04 | **0.01** | 0.02 |

### A.3.5 Additional Effect of Training Paradigm

Table A5 provides tabular results shown in Figure 3, comparing MIA efficacy against R-50 models fine-tuned on CIFAR-10 using three different parameterization strategies (Head-only, FiLM, and ALL) across four shots ($S = 16, 64, 256, 1024$).

Table A5: MIA efficacy against R-50 fine-tuned on CIFAR-10 across 3 parameterization strategies (Head-only, FiLM, and ALL) and 4 different $S$ (shots). Values represent medians with interquartile ranges (IQRs). Results are averaged over 5 repeats with 1 target model in each repeat.

| Shots | Attack | Parameterization | TPR@FPR=0.1 | | TPR@FPR=0.01 | | TPR@FPR=0.001 | |
|---|---|---|---|---|---|---|---|---|
| | | | Median | IQR | Median | IQR | Median | IQR |
| $S = 16$ | LiRA ($M = 64$) | Head-only | **0.89** | 0.20 | 0.51 | 0.37 | 0.51 | 0.37 |
| | | FiLM | **0.90** | 0.07 | **0.73** | 0.12 | **0.73** | 0.12 |
| | | ALL | **1.00** | 0.00 | **0.98** | 0.02 | **0.98** | 0.02 |
| | RMIA ($M = 64$) | Head-only | 0.16 | 0.37 | 0.10 | 0.16 | 0.10 | 0.16 |
| | | FiLM | **0.90** | 0.11 | 0.67 | 0.15 | 0.67 | 0.15 |
| | | ALL | **1.00** | 0.00 | **0.95** | 0.04 | **0.95** | 0.04 |
| | ML-Leaks ($M = 1$) | Head-only | 0.39 | 0.23 | 0.03 | 0.04 | 0.00 | 0.00 |
| | | FiLM | 0.53 | 0.14 | 0.01 | 0.18 | 0.01 | 0.18 |
| | | ALL | **0.99** | 0.01 | 0.83 | 0.24 | 0.83 | 0.24 |
| | Trajectory-MIA ($M = 3$) | Head-only | 0.31 | 0.26 | 0.02 | 0.04 | 0.00 | 0.00 |
| | | FiLM | 0.33 | 0.06 | 0.01 | 0.01 | 0.01 | 0.01 |
| | | ALL | 0.51 | 0.25 | 0.11 | 0.03 | 0.11 | 0.03 |
| | | Head-only | 0.40 | 0.19 | 0.05 | 0.04 | 0.00 | 0.00 |

Table A5 – continued from previous page

| Shots | Attack | Parameterization | TPR@FPR=0.1 | | TPR@FPR=0.01 | | TPR@FPR=0.001 | |
|---|---|---|---|---|---|---|---|---|
| | | | Median | IQR | Median | IQR | Median | IQR |
| | Attack-P | FiLM | 0.53 | 0.14 | 0.01 | 0.16 | 0.00 | 0.02 |
| | | ALL | **0.99** | 0.01 | 0.60 | 0.17 | 0.06 | 0.02 |
| | QMIA | Head-only | 0.60 | 0.29 | 0.16 | 0.11 | 0.15 | 0.09 |
| | | FiLM | 0.56 | 0.04 | 0.08 | 0.05 | 0.06 | 0.08 |
| | | ALL | **0.99** | 0.00 | 0.88 | 0.09 | 0.86 | 0.10 |
| | LiRA ($M = 64$) | Head-only | **0.57** | 0.07 | **0.22** | 0.08 | **0.13** | 0.01 |
| | | FiLM | **0.55** | 0.07 | **0.26** | 0.10 | **0.15** | 0.09 |
| | | ALL | **0.94** | 0.03 | **0.72** | 0.06 | **0.42** | 0.20 |
| | RMIA ($M = 64$) | Head-only | 0.12 | 0.19 | 0.05 | 0.09 | 0.03 | 0.08 |
| | | FiLM | 0.41 | 0.07 | 0.24 | 0.02 | 0.14 | 0.04 |
| | | ALL | 0.70 | 0.05 | 0.42 | 0.10 | 0.24 | 0.12 |
| $S = 64$ | ML-Leaks ($M = 1$) | Head-only | 0.17 | 0.04 | 0.03 | 0.01 | 0.00 | 0.00 |
| | | FiLM | 0.12 | 0.04 | 0.02 | 0.01 | 0.00 | 0.01 |
| | | ALL | 0.39 | 0.17 | 0.06 | 0.01 | 0.02 | 0.02 |
| | Trajectory-MIA ($M = 3$) | Head-only | 0.27 | 0.39 | 0.02 | 0.21 | 0.00 | 0.17 |
| | | FiLM | 0.15 | 0.03 | 0.02 | 0.02 | 0.01 | 0.01 |
| | | ALL | NaN | NaN | NaN | NaN | NaN | NaN |
| | Attack-P | Head-only | 0.16 | 0.03 | 0.02 | 0.02 | 0.00 | 0.00 |
| | | FiLM | 0.14 | 0.06 | 0.02 | 0.01 | 0.00 | 0.00 |
| | | ALL | 0.41 | 0.07 | 0.08 | 0.02 | 0.01 | 0.01 |
| | QMIA | Head-only | 0.26 | 0.03 | 0.04 | 0.01 | 0.02 | 0.01 |
| | | FiLM | 0.17 | 0.05 | 0.03 | 0.02 | 0.01 | 0.00 |
| | | ALL | 0.42 | 0.08 | 0.08 | 0.21 | 0.02 | 0.15 |
| | LiRA ($M = 64$) | Head-only | 0.25 | 0.06 | 0.10 | 0.04 | 0.06 | 0.02 |
| | | FiLM | **0.29** | 0.03 | **0.14** | 0.01 | **0.08** | 0.01 |
| | | ALL | **0.44** | 0.11 | **0.20** | 0.05 | **0.13** | 0.03 |
| | RMIA ($M = 64$) | Head-only | 0.11 | 0.07 | 0.06 | 0.04 | 0.03 | 0.03 |
| | | FiLM | 0.22 | 0.03 | 0.11 | 0.00 | 0.06 | 0.01 |
| | | ALL | 0.23 | 0.06 | 0.15 | 0.01 | 0.07 | 0.01 |
| $S = 256$ | ML-Leaks ($M = 1$) | Head-only | 0.11 | 0.01 | 0.01 | 0.00 | 0.00 | 0.00 |
| | | FiLM | 0.11 | 0.01 | 0.01 | 0.00 | 0.00 | 0.00 |
| | | ALL | 0.15 | 0.02 | 0.02 | 0.00 | 0.00 | 0.00 |
| | Trajectory-MIA ($M = 3$) | Head-only | 0.17 | 0.18 | 0.06 | 0.10 | 0.02 | 0.04 |
| | | FiLM | 0.16 | 0.04 | 0.04 | 0.02 | 0.02 | 0.01 |
| | | ALL | 0.12 | 0.01 | 0.01 | 0.01 | 0.00 | 0.00 |
| | Attack-P | Head-only | 0.11 | 0.01 | 0.01 | 0.01 | 0.00 | 0.00 |
| | | FiLM | 0.11 | 0.01 | 0.01 | 0.00 | 0.00 | 0.00 |
| | | ALL | 0.17 | 0.02 | 0.01 | 0.00 | 0.00 | 0.00 |
| | QMIA | Head-only | 0.19 | 0.06 | 0.02 | 0.02 | 0.01 | 0.00 |
| | | FiLM | 0.13 | 0.00 | 0.02 | 0.00 | 0.00 | 0.00 |
| | | ALL | 0.17 | 0.03 | 0.04 | 0.03 | 0.01 | 0.01 |
| | LiRA ($M = 64$) | Head-only | **0.17** | 0.01 | **0.05** | 0.01 | **0.03** | 0.01 |
| | | FiLM | **0.18** | 0.01 | **0.06** | 0.01 | **0.03** | 0.01 |
| | | ALL | **0.33** | 0.04 | **0.14** | 0.01 | **0.07** | 0.01 |
| | RMIA ($M = 64$) | Head-only | 0.12 | 0.10 | 0.04 | 0.01 | 0.02 | 0.01 |
| | | FiLM | **0.17** | 0.01 | **0.06** | 0.00 | **0.03** | 0.01 |
| | | ALL | 0.23 | 0.04 | **0.13** | 0.03 | 0.05 | 0.00 |

**Table A5 – continued from previous page**

| Shots | Attack | Parameterization | TPR@FPR=0.1 | | TPR@FPR=0.01 | | TPR@FPR=0.001 | |
|---|---|---|---|---|---|---|---|---|
| | | | Median | IQR | Median | IQR | Median | IQR |
| | ML-Leaks ($M = 1$) | Head-only | 0.10 | 0.01 | 0.01 | 0.00 | 0.00 | 0.00 |
| | | FiLM | 0.10 | 0.01 | 0.01 | 0.00 | 0.00 | 0.00 |
| | | ALL | 0.11 | 0.02 | 0.01 | 0.00 | 0.00 | 0.00 |
| | Trajectory-MIA ($M = 3$) | Head-only | 0.16 | 0.01 | **0.05** | 0.01 | 0.02 | 0.01 |
| | | FiLM | 0.12 | 0.01 | 0.02 | 0.00 | 0.00 | 0.00 |
| $S = 1024$ | | ALL | 0.15 | 0.06 | 0.03 | 0.02 | 0.01 | 0.01 |
| | Attack-P | Head-only | 0.10 | 0.00 | 0.01 | 0.00 | 0.00 | 0.00 |
| | | FiLM | 0.10 | 0.00 | 0.01 | 0.00 | 0.00 | 0.00 |
| | | ALL | 0.11 | 0.01 | 0.01 | 0.00 | 0.00 | 0.00 |
| | QMIA | Head-only | 0.13 | 0.02 | 0.03 | 0.01 | 0.00 | 0.00 |
| | | FiLM | 0.12 | 0.01 | 0.01 | 0.00 | 0.00 | 0.00 |
| | | ALL | 0.22 | 0.09 | **0.12** | 0.05 | **0.06** | 0.03 |

### A.3.6 Additional Effect of Number of Shadow Models

Table A6 provides tabular results investigating the relationship between $M$ (the number of shadow models) and MIA efficacy for LiRA and RMIA. The experimental setup employs each of $M + 1$ trained models as the target with the remaining $M$ models serving as shadow models, allowing for a robust evaluation of the effect of shadow model scaling on ViT-B/16 models with Head-only fine-tuned on CIFAR-10.

Table A6: MIA efficacy as a function of the number of shadow models ($M$) for LiRA and RMIA against ViT-B/16 model with Head-only fine-tuned on CIFAR-10. Values represent medians with interquartile ranges (IQRs). For each configuration, we train $M + 1$ models per repeat, using each model as the target while the remaining $M$ serve as shadow models. MIA efficacy is averaged over 5 repeats and across all $M + 1$ target models per repeat.

| Shots | # of Shadow Models | Attack | TPR@FPR=0.1 | | TPR@FPR=0.01 | | TPR@FPR=0.001 | |
|---|---|---|---|---|---|---|---|---|
| | | | Median | IQR | Median | IQR | Median | IQR |
| | $M = 4$ | LiRA | 0.38 | 0.12 | 0.06 | 0.02 | 0.05 | 0.02 |
| | | RMIA | 0.22 | 0.12 | 0.06 | 0.05 | 0.04 | 0.05 |
| | $M = 16$ | LiRA | **0.83** | 0.13 | 0.40 | 0.09 | 0.39 | 0.09 |
| | | RMIA | 0.19 | 0.12 | 0.03 | 0.06 | 0.03 | 0.06 |
| $S = 16$ | $M = 32$ | LiRA | **0.87** | 0.08 | **0.53** | 0.14 | **0.53** | 0.14 |
| | | RMIA | 0.17 | 0.10 | 0.08 | 0.06 | 0.08 | 0.06 |
| | $M = 64$ | LiRA | **0.89** | 0.22 | **0.51** | 0.45 | **0.51** | 0.45 |
| | | RMIA | 0.12 | 0.03 | 0.02 | 0.14 | 0.02 | 0.14 |
| | $M = 128$ | LiRA | **0.90** | 0.09 | **0.60** | 0.11 | **0.60** | 0.11 |
| | | RMIA | 0.16 | 0.08 | 0.05 | 0.04 | 0.05 | 0.04 |
| | $M = 256$ | LiRA | **0.90** | 0.08 | **0.59** | 0.12 | **0.59** | 0.12 |
| | | RMIA | 0.15 | 0.07 | 0.05 | 0.04 | 0.05 | 0.04 |
| | $M = 4$ | LiRA | 0.24 | 0.02 | 0.02 | 0.00 | 0.00 | 0.00 |
| | | RMIA | 0.29 | 0.09 | 0.12 | 0.06 | 0.06 | 0.02 |
| | $M = 16$ | LiRA | 0.52 | 0.05 | 0.18 | 0.04 | 0.08 | 0.01 |
| $S = 64$ | | RMIA | 0.23 | 0.15 | 0.11 | 0.09 | 0.08 | 0.07 |
| | $M = 32$ | LiRA | 0.55 | 0.05 | **0.25** | 0.01 | **0.14** | 0.01 |

Table A6 – continued from previous page

| Shots | # of Shadow Models | Attack | TPR@FPR=0.1 | | TPR@FPR=0.01 | | TPR@FPR=0.001 | |
|---|---|---|---|---|---|---|---|---|
| | | | Median | IQR | Median | IQR | Median | IQR |
| | | RMIA | 0.21 | 0.13 | 0.11 | 0.04 | 0.08 | 0.04 |
| | $M = 64$ | LiRA | **0.57** | 0.05 | 0.24 | 0.08 | **0.14** | 0.00 |
| | | RMIA | 0.27 | 0.19 | 0.12 | 0.12 | 0.09 | 0.07 |
| | $M = 128$ | LiRA | **0.59** | 0.01 | **0.27** | 0.02 | **0.17** | 0.03 |
| | | RMIA | 0.19 | 0.08 | 0.10 | 0.07 | 0.06 | 0.05 |
| | $M = 256$ | LiRA | **0.60** | 0.03 | **0.28** | 0.03 | **0.18** | 0.04 |
| | | RMIA | 0.18 | 0.09 | 0.10 | 0.05 | 0.07 | 0.05 |
| $S = 256$ | $M = 4$ | LiRA | 0.14 | 0.01 | 0.01 | 0.00 | 0.00 | 0.00 |
| | | RMIA | 0.16 | 0.04 | 0.06 | 0.03 | 0.03 | 0.02 |
| | $M = 16$ | LiRA | 0.23 | 0.04 | 0.06 | 0.01 | 0.02 | 0.00 |
| | | RMIA | 0.12 | 0.07 | 0.06 | 0.04 | 0.03 | 0.03 |
| | $M = 32$ | LiRA | **0.25** | 0.05 | **0.09** | 0.03 | 0.04 | 0.01 |
| | | RMIA | 0.11 | 0.05 | 0.05 | 0.03 | 0.03 | 0.02 |
| | $M = 64$ | LiRA | **0.28** | 0.05 | **0.11** | 0.04 | **0.07** | 0.02 |
| | | RMIA | 0.13 | 0.02 | 0.07 | 0.04 | 0.03 | 0.03 |
| | $M = 128$ | LiRA | **0.28** | 0.05 | **0.12** | 0.03 | **0.06** | 0.02 |
| | | RMIA | 0.10 | 0.05 | 0.05 | 0.03 | 0.03 | 0.03 |
| | $M = 256$ | LiRA | **0.29** | 0.05 | **0.12** | 0.03 | **0.07** | 0.02 |
| | | RMIA | 0.11 | 0.05 | 0.05 | 0.03 | 0.03 | 0.03 |
| $S = 1024$ | $M = 4$ | LiRA | 0.11 | 0.01 | 0.01 | 0.00 | 0.00 | 0.00 |
| | | RMIA | 0.11 | 0.05 | 0.04 | 0.01 | 0.02 | 0.01 |
| | $M = 16$ | LiRA | 0.15 | 0.00 | 0.03 | 0.00 | 0.01 | 0.00 |
| | | RMIA | 0.10 | 0.03 | 0.04 | 0.00 | 0.02 | 0.01 |
| | $M = 32$ | LiRA | 0.16 | 0.01 | 0.04 | 0.00 | 0.02 | 0.00 |
| | | RMIA | 0.09 | 0.05 | 0.04 | 0.01 | 0.02 | 0.01 |
| | $M = 64$ | LiRA | **0.18** | 0.01 | 0.05 | 0.00 | **0.03** | 0.01 |
| | | RMIA | 0.12 | 0.08 | 0.04 | 0.00 | 0.02 | 0.00 |
| | $M = 128$ | LiRA | **0.18** | 0.01 | **0.06** | 0.00 | **0.03** | 0.01 |
| | | RMIA | 0.09 | 0.04 | 0.04 | 0.00 | 0.02 | 0.01 |
| | $M = 256$ | LiRA | **0.18** | 0.01 | **0.06** | 0.00 | **0.03** | 0.00 |
| | | RMIA | 0.09 | 0.05 | 0.04 | 0.00 | 0.02 | 0.01 |

### A.3.7 Additional Effect of Using Data Augmentation During Fine-Tuning

Table A7 provides tabular results shown in Figure 5, examining the effect of data augmentation on MIA efficacy for ViT-B/16 models fine-tuned on CIFAR-10. The table compares two augmentation methods (+ Mirror and + Shift) across 4 different shot values ($S = 16, 64, 256, 1024$).

Table A7: Impact of data augmentation on MIA efficacy across $S$ (shots) for ViT-B/16 models Head-only fine-tuned on CIFAR-10 *with data augmentations*. We compare 2 augmentation strategies: *+ Mirror* (where original image plus a horizontally flipped copy of it are used to train the target model) and *+ Shift* (where horizontally flipping and/or $\pm 1$-pixel shifts are applied to the original image). Values represent medians with interquartile ranges (IQRs). Results are averaged over 5 repeats and we use $M + 1$ target models ($M = 64$) per repeat.

| Shots | Attack | Augmentation | TPR@FPR=0.1 | | TPR@FPR=0.01 | | TPR@FPR=0.001 | |
|---|---|---|---|---|---|---|---|---|
| | | | Median | IQR | Median | IQR | Median | IQR |
| $S = 16$ | LiRA ($M = 64$) | No augmentation | **0.89** | 0.03 | **0.55** | 0.07 | **0.55** | 0.07 |
| | | + Mirror | **0.89** | 0.03 | **0.55** | 0.07 | **0.55** | 0.07 |
| | | + Shift | **0.91** | 0.03 | **0.58** | 0.07 | **0.58** | 0.07 |
| | RMIA ($M = 64$) | No augmentation | 0.28 | 0.40 | 0.10 | 0.23 | 0.10 | 0.23 |
| | | + Mirror | 0.26 | 0.40 | 0.09 | 0.20 | 0.09 | 0.20 |
| | | + Shift | 0.30 | 0.38 | 0.10 | 0.20 | 0.10 | 0.20 |
| $S = 64$ | LiRA ($M = 64$) | No augmentation | **0.42** | 0.04 | **0.15** | 0.02 | **0.08** | 0.00 |
| | | + Mirror | **0.42** | 0.04 | **0.15** | 0.02 | **0.08** | 0.00 |
| | | + Shift | **0.43** | 0.04 | **0.16** | 0.03 | **0.09** | 0.01 |
| | RMIA ($M = 64$) | No augmentation | 0.13 | 0.02 | 0.04 | 0.04 | 0.02 | 0.04 |
| | | + Mirror | 0.12 | 0.02 | 0.04 | 0.03 | 0.02 | 0.03 |
| | | + Shift | 0.13 | 0.04 | 0.04 | 0.03 | 0.02 | 0.03 |
| $S = 256$ | LiRA ($M = 64$) | No augmentation | **0.19** | 0.02 | **0.06** | 0.01 | **0.03** | 0.01 |
| | | + Mirror | **0.19** | 0.02 | **0.06** | 0.01 | **0.03** | 0.01 |
| | | + Shift | **0.20** | 0.02 | **0.06** | 0.01 | **0.03** | 0.01 |
| | RMIA ($M = 64$) | No augmentation | 0.10 | 0.01 | 0.03 | 0.01 | 0.00 | 0.00 |
| | | + Mirror | 0.09 | 0.01 | 0.03 | 0.01 | 0.01 | 0.00 |
| | | + Shift | 0.10 | 0.01 | 0.03 | 0.01 | 0.01 | 0.01 |
| $S = 1024$ | LiRA ($M = 64$) | No augmentation | 0.15 | 0.00 | 0.04 | 0.00 | 0.02 | 0.00 |
| | | + Mirror | 0.15 | 0.00 | 0.04 | 0.00 | 0.02 | 0.00 |
| | | + Shift | **0.16** | 0.00 | 0.04 | 0.00 | 0.02 | 0.00 |
| | RMIA ($M = 64$) | No augmentation | 0.12 | 0.01 | 0.04 | 0.00 | 0.02 | 0.00 |
| | | + Mirror | 0.11 | 0.01 | 0.04 | 0.00 | 0.02 | 0.00 |
| | | + Shift | 0.11 | 0.01 | 0.04 | 0.00 | 0.02 | 0.00 |

