# OpenReview forum: "Empirical Comparison of Membership Inference Attacks in Deep Transfer Learning"
_TMLR — Accepted by TMLR_

### Review · Reviewer_uhSE · 2025-07-20

**Summary Of Contributions:**

In this paper, the authors present extensive evaluations of existing membership inference attacks on transfer learning settings, mainly showcasing that attack performance generally decreases as the fine-tuning dataset size increases. However, the authors also find that this general rule of thumb is not always true, and that there exist attacks/setup configurations where increasing fine-tuning dataset size does not reduce performance (IHA on binary dataset). The authors also find that across the attacks they evaluated, model training choices like head-only, FiLM strategies, and data augmentations do not make a significant difference in attack performance, while attack parameters, like # of shadow models have more affect.

**Audience:**

Yes

**Claims And Evidence:**

Yes

**Requested Changes:**

**Low-shot model performance clarification**

In figures 2, 3, 4, it is implied that the authors trained models using at the lowest, only 4 samples per class in the fine-tuning set. Is the fine-tuned model able to learn the fine-tuning datasets with that little amount of data? Can the authors report test scores of their fine-tuned models?

**Missing ROC Curves**

In section 4, it is indicated that the authors would report ROC curves in addition to their other metrics. However I do not see any ROC curves in the manuscript.

**Full parameter fine-tuning**

A key missing component of this paper is that full-parameter fine-tuning does not appear to be considered. Can the authors explain why? or add results for full-parameter fine-tuning?

**Strengths And Weaknesses:**

**Strengths**

- results are evaluated across a very comprehensive set of MIA
- confirms prior literature with strong empirical results

**Weaknesses**

- main results only confirm existing knowledge
- missing full-parameter fine-tuning (see requested changes)

---

> ### Author Response · Authors · 2025-08-22
> **Response to Reviewer uhSE**
>
> Thanks for your review. We have uploaded a revision and highlighted changes to the original submission in blue.
>
> > **Low-shot model performance clarification**
>
> We have added plots depicting the training and test accuracies of the fine-tuned models trained with varying no. of shots in Appendix A.3.1. We have included plots depicting accuracies for the following experiments:
> [Figure A3, Page 16] ViT-B/16 Head-Only fine-tuned models for different datasets (CIFAR-10, CIFAR-100, PatchCamelron), and
> [Figure A4, Page 16] R-50 model training with CIFAR-10 using different parameterization schemes (Head-Only, FiLM, ALL).
>
> > **Missing ROC Curves**
>
> We have added plots for ROC curves in Appendix A.3.2 [Figures A5,A6,A7], corresponding to the training setting shown in Figure 2, where we perform our most comprehensive comparison of attacks. Hope that addresses the reviewer’s concern.
>
> > **Full parameter fine-tuning**
>
> We have added results for full-parameter fine-tuning (for the R-50 model trained using CIFAR-10) in Figure 3, Page 8 (and associated training and test accuracies for full-parameter fine-tuning in Figure A4, Page 16).

---

### Review · Reviewer_82yg · 2025-07-21

**Summary Of Contributions:**

This paper studies the effectiveness of membership inference attacks under transfer learning setting. This work consider score-based MIAs including both shadow-model-based attacks and shadow-model-free attacks. This worked evaluated several MIAs including LiRA, RMIA,  ML-Leaks, Trajectory-MIA, Attack-P, QMIA, and IHA for two transfer learning settings: head-only and FiLM adapters. This work provides the  empirical evaluation of these MIAs on CIFAR10, CIFAR100 and PatchCamelyon datasets, and provides analysis for the factors that influence MIA efficacy including number of shots, FiLM vs Head-only,  number of shadow models. effect of using data augmentation.

**Audience:**

Yes

**Broader Impact Concerns:**

NA.

**Claims And Evidence:**

No

**Requested Changes:**

Please address my concerns in weaknesses.

**Strengths And Weaknesses:**

- strengths:
1. This work investigates a problem with practical implications, as transfer learning is widely deployed in practice. The paper is well-written and easy to follow.

2. This work performs hyper-parameter optimization to identify the optimal set of hyper-parameters to train the target model for a fair comparison.

3. This work provides analysis for the factors that influence MIA efficacy to understand the properties of different attacks.

- weaknesses:
1. It is nice to have Table 1 to demonstrate the target model access. In addition, it would also helpful to include the data access assumption. For example, according to Suri et al.2024, "IHA assumes knowledge of all the other n − 1 records in an n-sized dataset" and therefore presents IHA as privacy auditing tool. It would be helpful to also discuss the data access of these attacks.

2. A key characteristic of transfer learning is the domain shift between pre-training and fine-tuning. It would be helpful to perform the analysis in Section 5.2, 5.3, 5.4 for task with more domain shift such as PatchCamelyon to understand if there is any discrepancy between ID setting (the CIFAR10 in the current manuscript) and OOD setting.

3. For the data augmentation result, the representation learning in pre-training aims to learn the same representation for the same image with different augmentations, This could be the reason that the data augmentations cannot improve the MIAs in transfer learning. On the other hand, this also indicates that different augmentations in transfer learning does not bring additional information gain and might not improve the utility. I therefore have concerns about the discussion in Section 6 "practitioners can leverage augmentation techniques to improve model utility without substantially compromising its privacy." that needs further justification.

---

> ### Author Response · Authors · 2025-08-22
> **Response to Reviewer 82yg**
>
> Thanks for your review. We have uploaded a revision and highlighted changes to the original submission in blue.
>
> > It is nice to have Table 1 to demonstrate the target model access. In addition, it would also helpful to include the data access assumption. For example, according to Suri et al.2024, "IHA assumes knowledge of all the other n − 1 records in an n-sized dataset" and therefore presents IHA as privacy auditing tool. It would be helpful to also discuss the data access of these attacks.
>
> We have made suggested changes to the paper to clarify the white-box data access condition associated with IHA.
>
> “IHA relies on 2 key assumptions: for a given target sample, it assumes knowledge of all $n-1$ records in a $n$-sized dataset except for the target sample, and white-box access to the target model's parameters (Section 3.3 in (Suri et al. (2024)). In contrast, other attacks listed in Table 1 qualify as Black-box MIAs, which do not rely on such knowledge as a prerequisite. In Black-box MIAs, the attacker can only access the target model's response on a given target sample.”
>
> In addition to that, we have added a column in Table 1 named “Training Data Access” which informs the reader(s) that IHA assumes the attacker has knowledge of all but the target record in the training data.
>
> >... It would be helpful to perform the analysis in Section 5.2, 5.3, 5.4 for task with more domain shift such as PatchCamelyon to understand if there is any discrepancy between ID setting (the CIFAR10 in the current manuscript) and OOD setting.
>
> We have performed the analyses from Sections 5.2 [Figure 4, Page 9] and 5.3 [Figure A8, Page 20) using PatchCamelyon as the OOD dataset. Both analyses demonstrate consistent trends between the ID setting (CIFAR-10) and OOD setting (PatchCamelyon). Note that Trajectory-MIA results are omitted from Figure A8 as we encountered CUDA memory limitations during the distillation process required for this attack when fine-tuning with FiLM parameterization.
> We are toning down our claims for Section 5.4 (Effect of Using Data Augmentation During Fine-Tuning) because the only computationally viable setting of fine-tuning the final linear layer (Head-only) is so weak that augmentation does not have a significant impact on MIA efficacy. Thus, we saw no added value in running those experiments for PatchCamelyon, which would justify their associated cost.
>
> >  … I therefore have concerns about the discussion in Section 6 "practitioners can leverage augmentation techniques to improve model utility without substantially compromising its privacy." that needs further justification.
>
> We agree with the reviewer that, without an appropriate demonstration of augmentation being useful to improve model utility in the transfer learning setting, the highlighted claim would be unjustified. Thus, we have qualified our claim per the reviewer’s suggestion. The Discussion section now says,
> “However, in Section 5.4 we observe no significant improvement in MIA efficacy due to augmentation against models with the last linear layer subject to fine-tuning.“
> Furthermore, we have added a subplot in Figure 5 depicting the test accuracy of models trained with and without augmentations. From the figure, we can discern that for the Head-Only fine-tuning, training with or without augmentations does not significantly impact the MIA efficacy or the test accuracy of the models.

---

> > ### Comment · Reviewer_82yg · 2025-09-08
> >
> > Thank you for your response! The responses have addressed most of my concerns.
> >
> > I just have a minor follow-up comment:
> >
> > It seems that some plots in Figure 3 (Trajectory-MIA) and Figure A8 (ML-leaks) are not the line charts as others (maybe some results are missing, though I agree that the trend is consistent with the statements in the paper).

---

> > > ### Author Response · Authors · 2025-09-10
> > >
> > > Thanks for your follow-up comment! We have added Table A1 (corresponding to Figure A8) and updated Table A5 (corresponding to Figure 3) with tabular data from ALL fine-tuning experiments.
> > >
> > > Most of the missing data points in these figures are due to the poor performance of Trajectory-MIA and ML-Leaks on certain configurations. These values are highlighted in blue in the tables.
> > >
> > > The one exception is the missing value for Trajectory-MIA with ALL fine-tuning on PatchCamelyon at shots $S=64$, which is due to out-of-memory (OOM) issues: the batch size determined by hyperparameter optimization was too large to run the model distillation step of the attack for the target and shadow models within our computational resources. We have also made a point to mention this in the updated captions for Figure 3 and A8.

---

### Review · Reviewer_3qR7 · 2025-07-22

**Summary Of Contributions:**

The authors compare various popular practical membership inference attacks, as well as one white-box inverse Hessian attack in the fine-tuning setting. The major results are consistent with existing literature, while they also found different behavior of inverse Hessian attack compared to the black-box MIAs.

**Audience:**

Yes

**Claims And Evidence:**

Yes

**Requested Changes:**

I would suggest the authors consider the following changes.
- Make the code publicly available. I will be more inclined to support the acceptance of this work if the code is open-sourced.
- Either clarify what the unique findings are in the transfer learning setting compared to the standard setup. Or the authors should at least explain what should be taken care of when considering the transfer learning setting compared to the standard setup. If none of these can be justified, I would suggest removing the emphasis on transfer learning.

**Strengths And Weaknesses:**

Strengths:
- Solid and comprehensive empirical comparison of popular practical MIAs.
- The finding on the comparison to white-box attack (IHA) is interesting.

Weaknesses:
- I found the main contribution of this work to be the solid experiment in comparing MIAs. Yet this contribution is significantly diminished as the authors do not provide their code.
- The major findings are known to the literature, and the findings also do not seem to have strong relations to transfer learning settings.

### Detail comments
I feel this is a solid empirical comparison of popular and practical MIA approaches in the literature. I am happy to see the authors include not only classical approaches (i.e., LiRA), but also SOTA methods such as RMIA. The experiment seems comprehensive, and the corresponding design is done nicely. While the major findings are aligned and known to the literature, I still think it would be a nice addition to the literature.

However, several drawbacks prevent me from supporting this work. First, I feel the major contribution of this work is its nice experiment design and setup. This contribution is diminished due to the fact that the authors do not provide their code. Another drawback is that the work does not have strong connections to transfer learning settings, which is one main selling point of the work and thus a bit misleading.

---

> ### Author Response · Authors · 2025-08-22
> **Response to Reviewer 3qR7**
>
> Thanks for your review. We have uploaded a revision and highlighted changes to the original submission in blue.
>
> > Make the code publicly available. I will be more inclined to support the acceptance of this work if the code is open-sourced.
>
> The (anonymised) code for our experiments is available at: https://anonymous.4open.science/r/benchmark-mia-tl-E1DE
>
> > Either clarify what the unique findings are in the transfer learning setting compared to the standard setup. Or the authors should at least explain what should be taken care of when considering the transfer learning setting compared to the standard setup. If none of these can be justified, I would suggest removing the emphasis on transfer learning.
> We understand the Reviewer’s concern about the lack of clarity associated with the transfer learning setting. We chose to focus on transfer learning in our paper because it is the most realistic approach to obtaining high-utility private deep learning models, and it is important to assess the vulnerability of models trained using transfer learning, given that it differs quite significantly from models trained from scratch (as shown in Figure 1).
>
> We have amended the Limitations [Page 10] to say, “Another limitation of our paper is that we do not offer a detailed comparison between the performance of different MIAs against models trained with fine-tuning and from-scratch beyond Figure 1. This is because running these experiments for models trained from scratch would be computationally expensive.”

---

### Decision · Action_Editor_RJHo · 2025-09-20

**Recommendation:** Accept as is

**Additional Comments:**

The paper provides a solid and comprehensive empirical comparison of popular practical MIAs. The experimental study is well executed and thorough. During the review process, the three reviewers raised several issues which were mostly addressed by the authors who performed additional experiments, added figures and released their code. One concern remained about whether the "transfer learning" setting should instead be interpreted more as fine-tuning.

My opinion is that the merits of the paper clearly overcome the concern mentioned above. In fact, I view the response by the authors on the matter and the paragraph added in the 'Limitations' (beginning of page 11) as sufficient. This empirical comparison will be of interest to TMLR's audience and I find that the paper fulfills all of TMLR's criteria for acceptance. As such, I recommend acceptance.

**Audience:**

Yes

**Audience Explanation:**

The paper provides a comprehensive study of membership inference attacks in transfer learning, a topic of clear relevance to TMLR's audience.

**Claims And Evidence:**

Yes

**Claims Explanation:**

The experiments are reproducible and justified (code has also been released, which was well appreciated); the claims are clear and justifiable.